# Structural basis for importin alpha 3 specificity of W proteins in Hendra and Nipah viruses

Kate M. Smith[1], Sofiya Tsimbalyuk[1], Megan R. Edwards[2], Emily M. Cross[1], Jyoti Batra[2],
Tatiana P. Soares da Costa[3], David Aragão [4], Christopher F. Basler [2] & Jade K. Forwood[1]

Seven human isoforms of importin α mediate nuclear import of cargo in a tissue- and isoform-specific manner. How nuclear import adaptors differentially interact with cargo harbouring the same nuclear localisation signal (NLS) remains poorly understood, as the NLS recognition region is highly conserved. Here, we provide a structural basis for the nuclear import specificity of W proteins in Hendra and Nipah viruses. We determine the structural interfaces of these cargo bound to importin α1 and α3, identifying a 2.4-fold more extensive interface and > 50-fold higher binding affinity for importin α3. Through the design of importin α1 and α3 chimeric and mutant proteins, together with structures of cargo-free importin α1 and α3 isoforms, we establish that the molecular basis of specificity resides in the differential positioning of the armadillo repeats 7 and 8. Overall, our study provides mechanistic insights into a range of important nucleocytoplasmic transport processes reliant on isoform adaptor specificity.

[1] School of Biomedical Sciences, Charles Sturt University, Wagga Wagga, NSW 2678, Australia. [2] Center for Microbial Pathogenesis, Institute for Biomedical Sciences, Georgia State University, Atlanta, GA 30303, USA. [3] Department of Biochemistry and Genetics, La Trobe Institute for Molecular Science, La Trobe University, Melbourne, VIC 3086, Australia. [4] Australian Synchrotron, Australian Nuclear Science and Technology Organisation, 800 Blackburn Road, Clayton, VIC 3168, Australia. These authors contributed equally: Kate M. Smith, Sofiya Tsimbalyuk, Megan R. Edwards. These authors jointly supervised this work: Christopher F. Basler, Jade K. Forwood. Correspondence and requests for materials should be addressed to C.F.B. (email: cbasler@gsu.edu) or to J.K.F. (email: jforwood@csu.edu.au)

Active transport of proteins from the cytoplasm to the nucleus is mediated by a family of nuclear transport receptors known as importins (or karyopherins), together with a number of ancillary proteins including nucleoporins and Ran[1–3]. The classical nuclear import pathway is best understood, and is initiated by recognition of proteins that contain a classical nuclear localisation sequence (NLS) by importin α[4]. This complex is transported through the nuclear pore complex by importin β, involving interactions with FG repeat regions contained within nucleoporin proteins[5,6]. Once the complex has traversed the nuclear envelope, RanGTP dissociates the complex, and the import receptors are recycled back to the cytoplasm to perform further rounds of transport[7–10]. Importin α is constructed from an N-terminal importin β-binding (IBB) domain and a C-terminal NLS binding domain featuring ten armadillo (ARM)-repeat motifs[11]. Most commonly, the cargo NLS binds on the concave site of the ARM repeats and involves interactions at either the major site, through ARM repeats 2–4 in the case of classical monopartite NLSs (e.g., SV40T-ag[12]), the minor site involving ARM repeats 6–8 (e.g., human phospholipid scramblase[13], TPX2[14]) or both the major and minor sites in the case of classical bipartite NLSs (e.g., nucleoplasmin[12]). Although this process has been well characterised for the importin α1 adaptor protein, many nuclear proteins exhibit specificity for other importin α isoforms, and the molecular basis for this specificity is understood poorly. Complicating our understanding of how importin α isoforms exhibit specificity, the seven importin α isoforms are highly conserved in the regions that mediate NLS binding. Establishing how nuclear cargo are recognised in an isoform-specific manner is important for understanding many key regulatory processes including cell differentiation, cancer and viral infection. For example, both RCC1 (the exchange factor of Ran that regulates the directionality of nuclear transport) and HIV-1 integrase (responsible for integrating the HIV-1 genome into the DNA of an infected cell) bind specifically to importin α3[15,16]. STAT1, a signalling molecule in the innate immune system response, binds specifically to the convex C-terminal surface of importin α5, α6 and α7[17,18]. The avian influenza PB2 viral polymerase subunit, which is a major virulence determinant, has isoform specificity for importin α3 in avian hosts and importin α7 in mammalian hosts, providing a kinetic advantage owing to lower importin α autoinhibition by the importin beta binding domain[19].

The W protein of Nipah virus (NiV) is recognised specifically by importin α3, which occurs through an NLS in the unique C-terminal domain[20,21], however, the basis of this specificity remains unclear. HeV and NiV are recently emergent, zoonotic pathogens of the *Henipavirus* genus in the *Paramyxoviridae* family. These viruses use bats of the *Pteropus* genus as reservoir hosts but cause infections in humans with a high rate of fatality (~ 60%)[22]. The viruses are non-segmented, negative-sense RNA viruses and encode six genes, five of which encode a single protein, whereas, the sixth gene, P, encodes four proteins: P, V, W and C. Of these, P, V and W share a common N-terminal domain, but differ within the C-terminal domain owing to frameshifting that results from insertion of non-template encoded nucleotides into P gene transcripts[23–25]. This results in proteins that have different functions and cellular locations, with only the W protein displaying steady-state nuclear localisation owing to a C-terminally located NLS[20,26]. In the context of NiV infection, the nonstructural W protein plays an important role in virulence[27–29]. It has been demonstrated to antagonise innate antiviral defences by blocking interferon-induced gene expression and by preventing expression of type I IFNs, with nuclear localisation shown to be important for the latter function[20,30]. A recent study has identified additional host targets of NiV W, including the PRP19 complex, for which the nuclear localisation of W appears to be required for the interaction[31].

Previous reports have demonstrated specificity of the importin α3 adaptor for the NiV W protein, however, like most cargo, the basis for this specificity is unknown[20,32]. Here, we establish key differences in the binding interface of W proteins with importin α1 and α3, providing insights into adaptor specificity. We use mutagenesis and chimeras of importin α1 and α3 to confirm the importance of these differences and show that the C-terminal ARM repeats 7 and 8 of importin α3 are important for specificity as their conformation in importin α3 allows for an extensive binding interface not possible in importin α1. These insights extend our understanding of adaptor specificity and establish how important nuclear cargo proteins are imported in an isoform-specific manner.

## Results

**Henipavirus W preferentially bind importin α3 and α4.** The W proteins of henipaviruses share a common N-terminal domain with P and V proteins, but have a unique NLS bearing C-terminal domain that is recognised preferentially by importins α3 and α4 (KPNA4 and KPNA3), rather than the better characterised importin α1 (KPNA2) (Fig. 1a and Supplementary Fig. 1A)[20]. To confirm this, and assess W binding against a more extensive range of importin α isoforms, we performed co-immunoprecipitation assays against respective importins and probed for the presence of HeV and NiV W. We found that both HeV and NiV W bound preferentially to importin α3 and α4, but not importin α1, α5, α6 or α7 (Fig. 1b and Supplementary Fig. 2). These importin αs extend across all subfamilies (Supplementary Fig. 1B), confirming that the W protein binds specifically to members of importin α subfamily-2.

To determine the binding affinities between importin α3 and Henipavirus W proteins, as well as confirm a direct interaction, the C-terminal NLS domain of the W protein from both HeV and NiV were cloned as a glutathione S-transferases (GST) fusion protein and assessed for importin α binding by isoforms from representative subfamilies. From enzyme-linked immunosorbent assay (ELISA) measurements, we found that importin α3 bound with high affinity to the W protein from both HeV and NiV (19.9 nM and 14.4 nM, respectively), whereas importin α1 and α7 bound with much lower affinity (1.4 μM and 1.5 μM, respectively, to HeV W; and 1.5 μM and 1.1 μM, respectively, to NiV W) (Fig. 1c; Supplementary Table 1). We found comparable binding using microscale thermophoresis (MST) assays, with importin α3 binding NiV W protein with high affinity (4.4 nM), whereas importin α1 and α7 displayed a much lower affinity (681 and 696 nM, respectively) (Fig. 1d; Supplementary Table 2). Together these data establish a direct, high-affinity interaction of the W proteins from henipaviruses for importin α3 compared with other importin α subfamilies.

**Structural basis for importin α isoform specificity to W.** To establish the mechanism and molecular basis of the isoform-specific differences, we used X-ray crystallography to determine the structural interface of both the high-affinity importin α3 interaction and the low-affinity importin α1 interaction. Structures were determined for importin α3 and α1 bound to both the HeV W and NiV W C-terminal NLS-bearing domain (Fig. 2 and Fig. 3, respectively). Crystals of the importin α1:HeV W complex had P2₁2₁2₁ symmetry and diffracted to 2.2 Å resolution. The asymmetric unit (ASU) contained one importin α1 chain bound to a single HeV W chain, in which residues 434–441 were bound to importin α1. The interaction was mediated by 15 hydrogen bonds, one salt bridge interaction, and buried a surface area of 683.5 Å² (see Table 1 for full

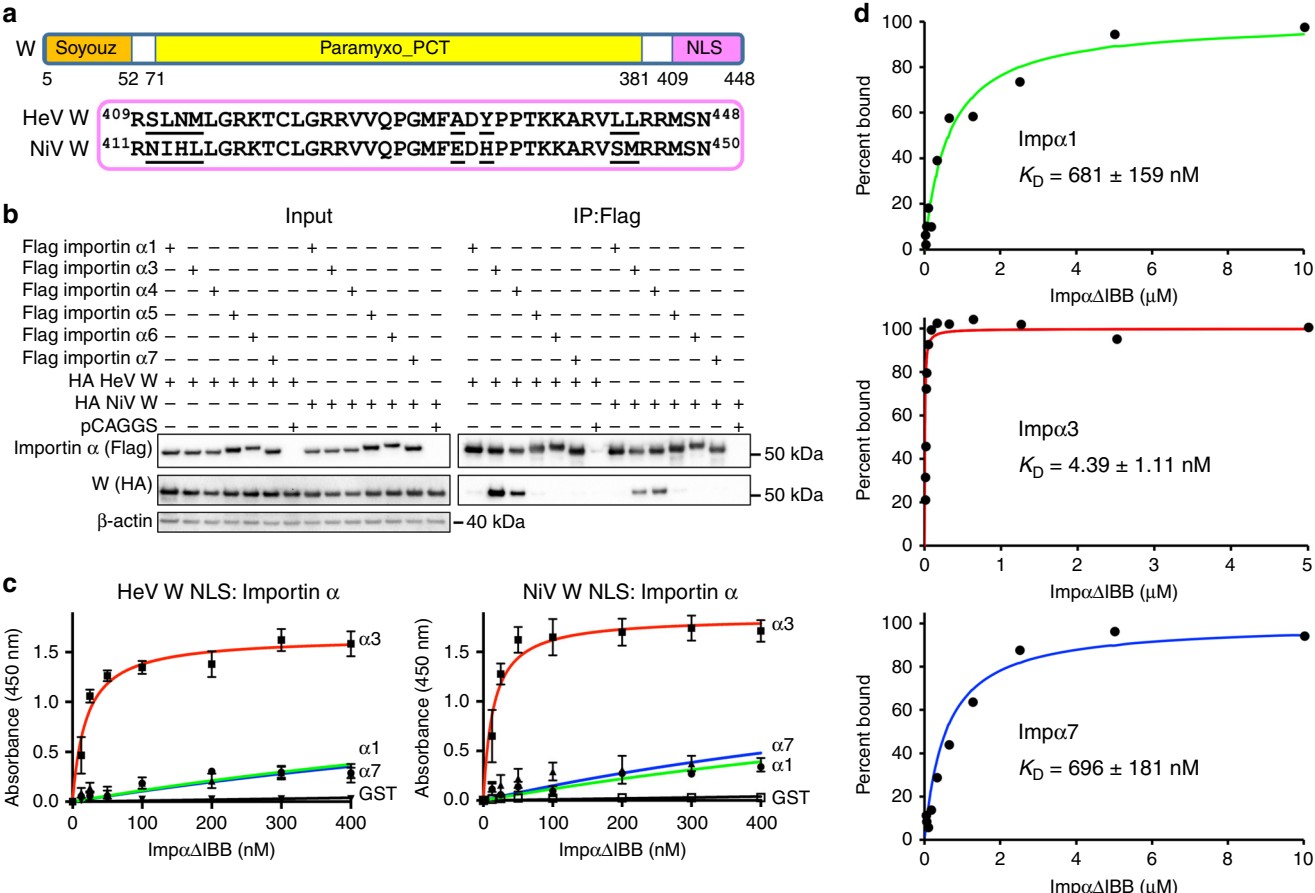

**Fig. 1** Binding of Henipavirus W proteins to importins. The NLS regions of Henipavirus W proteins bind with high affinity and specificity to the importin α2 subfamily containing importin α3 and α4. **a** The W proteins contain the Soyouz module moiety (Soyouz) and PCT disordered (Paramyxo_PCT) regions, which are conserved across paramyxovirus phosphoproteins. However, the W proteins of HeV and NiV W also possess a unique C terminus compared with other P gene products and contain an NLS that mediates translocation of W into the nucleus. **b** HeV W and NiV W interact with importin α3 and α4. Co-immunoprecipitation assay performed with Flag antibody on lysates of HEK293T cells expressing Flag-tagged importin α1, α3, α4, α5, α6, α7 and HA-tagged full length HeV W and NiV W, as indicated. Western blots were performed for HA and Flag. WCL, whole cell lysate; IP, immunoprecipitation. pCAGGS denotes empty vector control. **c** The NLS region of HeV W and NiV W interact with high affinity to importin α3. An ELISA was performed using GST-W (GST as a negative control) proteins coated on 96-well plates, and binding of 6xHis-tagged importin α1, α3, α7 assessed using an anti-6xHis HRP antibody. Error bars show the S.E.M for three replicates. **d** MST assay confirms high affinity binding of importin α3 to the NLS region of NiV W proteins, and comparatively lower affinity binding to importin α1 and α7

data collection and refinement statistics and Supplementary Table 3 for a detailed list of the interactions).

The structure of the importin α3:HeV W complex was determined in three different space groups (Table 2), with all crystals exhibiting highly similar structures (r.m.s.d < 0.35; Supplementary Fig. 3) and containing one HeV W NLS chain bound to a single importin α3 chain (the highest resolution structure of 1.6 Å will be used for further discussion and analysis). In contrast to the importin α1:HeV W complex, importin α3 bound a more extensive region of the HeV W C-terminal domain NLS (residues 419–444) and showed a more extensive array of hydrogen bonds (31 vs 15), seven salt bridge interactions (7 vs 1), and a greater buried surface area (1616.9 vs 683.5 Å$^2$); (see Supplementary Table 4 and Supplementary Table 5 for a detailed list of interactions and summary comparisons, respectively).

To test whether the same binding patterns were present in the NiV W protein, we examined the binding determinants of importin α:NiV W complexes using the same approach. We observed binding patterns that were very similar to those seen with HeV W for both importin α1 and α3 (Fig. 3). Crystals of the importin α1:

NiV W complex (that had P2$_1$2$_1$2$_1$ symmetry and diffracted to 2.1 Å resolution) bound residues 436–443 of the NiV W C-terminal domain. In comparison, the importin α3:NiV W complex had P12$_1$1 symmetry, diffracted to 2.3 Å resolution, and showed more extensive binding, with residues 421–446 bound to importin α3. The binding interface was also similar to HeV W, with the importin α1:NiV W complex mediated by 15 hydrogen bonds, one salt bridge interaction, and a buried surface area of 688.7 Å$^2$. The importin α3:NiV W interface was mediated through 31 hydrogen bonds, seven salt bridge interactions, and buried 1591.8 Å$^2$ of surface area. These results indicate that the interaction of the henipavirus W proteins with importins is highly conserved.

Establishing the structural interface between importin α1 and α3 in both HeV W and NiV W highlighted key differences in importin recognition and gave insights into isoform specificity. Notably, despite binding the same region of HeV W and NiV W, importin α3 exhibited a 2.4-fold more extensive interaction buried surface area than importin α1, with 16 additional hydrogen bonds, and six additional salt bridge interactions, consistent with its >50-fold higher affinity. Both W proteins

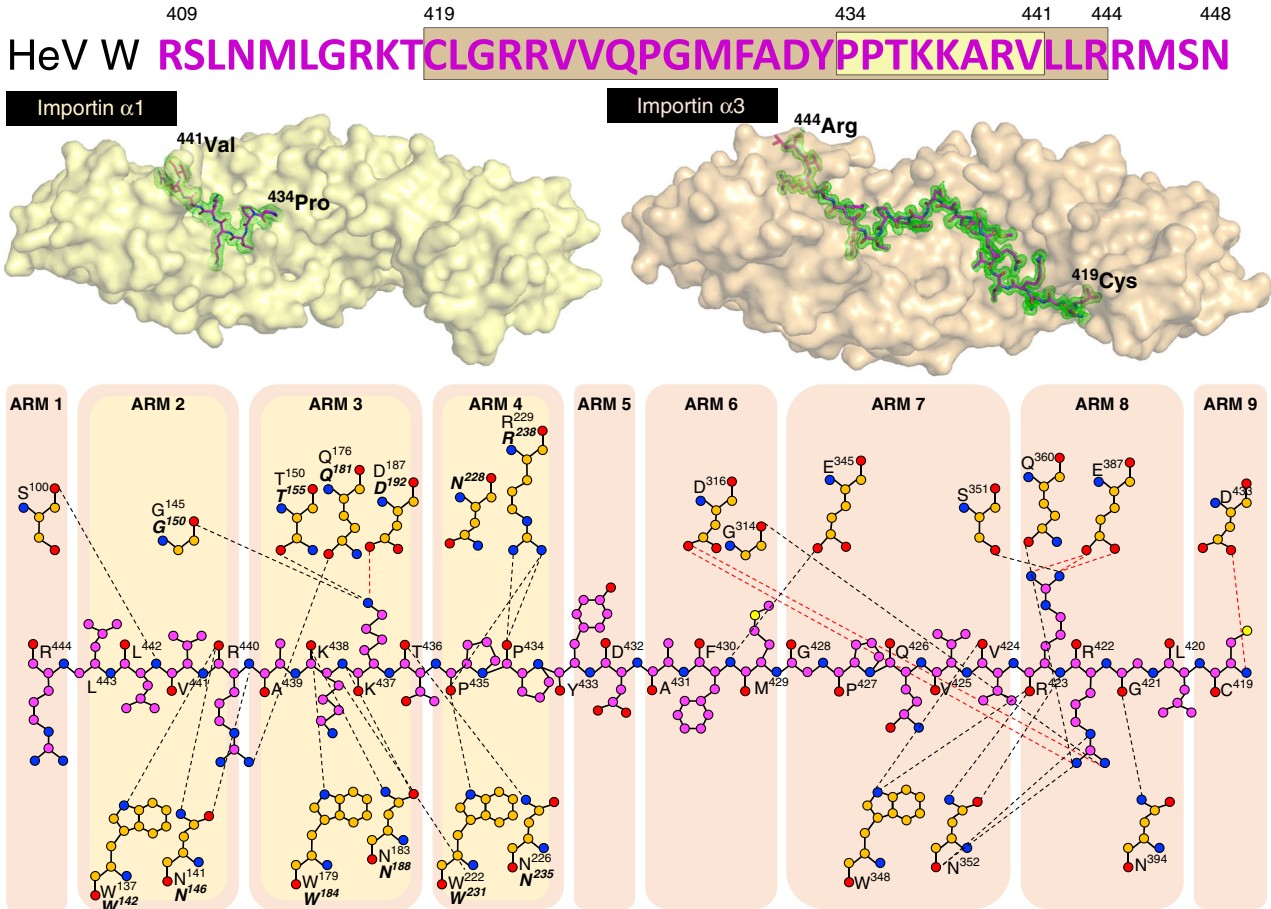

**Fig. 2** Structural comparisons and basis for high affinity interaction of HeV W with importin α3. Importin α1:HeV W and importin α3:HeV W structures were solved to 2.2 Å and 1.6 Å resolution, respectively. The importin α adaptors are shown in cartoon and surface representation (importin α1: bright yellow, importin α3: light orange) and HeV W NLS in stick representation (magenta: carbon atoms, red: oxygen atoms, blue: nitrogen atoms) with associated simulated annealing Fo-Fc omit maps of the NLSs contoured to 3σ in green. Schematics of the binding interface and specific interactions are shown below (magenta: carbon atoms, red: oxygen atoms, blue: nitrogen atoms), with hydrogen bond and salt bridge interactions depicted by dash lines (black and red respectively), and the partner interactions for importin α1 (bold italics) and importin α3 coloured as orange: carbon atoms, red: oxygen atoms, blue: nitrogen atoms. The location of these interactions in the ARM repeats highlight the greater interaction interface of the HeV W NLS for the importin α3 adaptor

interacted with the major binding site of importin α1 and α3, involving residues PPTKARV of W and ARM repeats 2–4 of importin in both structures. However, there were marked differences residing outside this region, with importin α3 ARM repeats 6–9 binding HeV W and NiV W residues 419–444 and 421–446, respectively (Figs. 2, 3). To probe the interaction of the importin α3:HeV W interface in greater detail, we designed mutations to disrupt key salt bridges at the major and minor sites of importin α3 (Fig. 4a). We found that mutations within the viral W proteins predicted to disrupt interactions at the minor site of importin α3 (specifically, R422A/R423A and R422D/R423D in the HeV W C-terminal domain and R424A/R425A and R424D/R425D in NiV W C-terminal domain) were no longer pulled down with importin α3 in our co-immunoprecipitation assay (Fig. 4b). Similarly, mutations predicted to disrupt interaction at the importin α3 major site, K437A/K438A and K437D/K438D in HeV W C-terminal domain, and K439A/K440A and K439D/K440D in the NiV W C-terminal domain, also failed to interact with importin α3. Consistent with these results, we found that mutations in both the major and minor site-binding regions of HeV W disrupted the affinity of interaction to importin α3 in ELISA (Fig. 4c), and resulted in reduced nuclear

accumulation (Fig. 4d, e). Overall, these results indicate that the binding of W to both the importin α major and minor sites is important for the high affinity interaction of henipavirus W to importin α3.

**ARMs 7 and 8 of importin α3 mediate isoform specificity**. Our structural analysis of the W protein bound to importin α1 and α3 showed that all binding determinants on importin α3 are conserved in importin α1 (Fig. 5a and Supplementary Fig. 4), suggesting that isoform specificity is not due to differences in the NLS-binding groove. This is consistent with previous work by Pumroy et al. [19]. To gain insights into the basis of isoform specificity, we superimposed the structures of importin α1:HeV W and importin α3:HeV W, and found the structures to be highly similar, with an r.m.s.d. of 1.1 Å across 411 residues (Fig. 5b), except for the positioning of ARM repeats 7 and 8. Notably, in importin α1, ARM repeats 7 and 8 are positioned closer to the inner surface of the NLS-binding interface, whereas in importin α3, these regions are positioned in a more open conformation. Because these ARM repeats provide a critical point of difference between the binding of importin α isoforms, we investigated why the W NLS region did not bind importin α1 in the extended

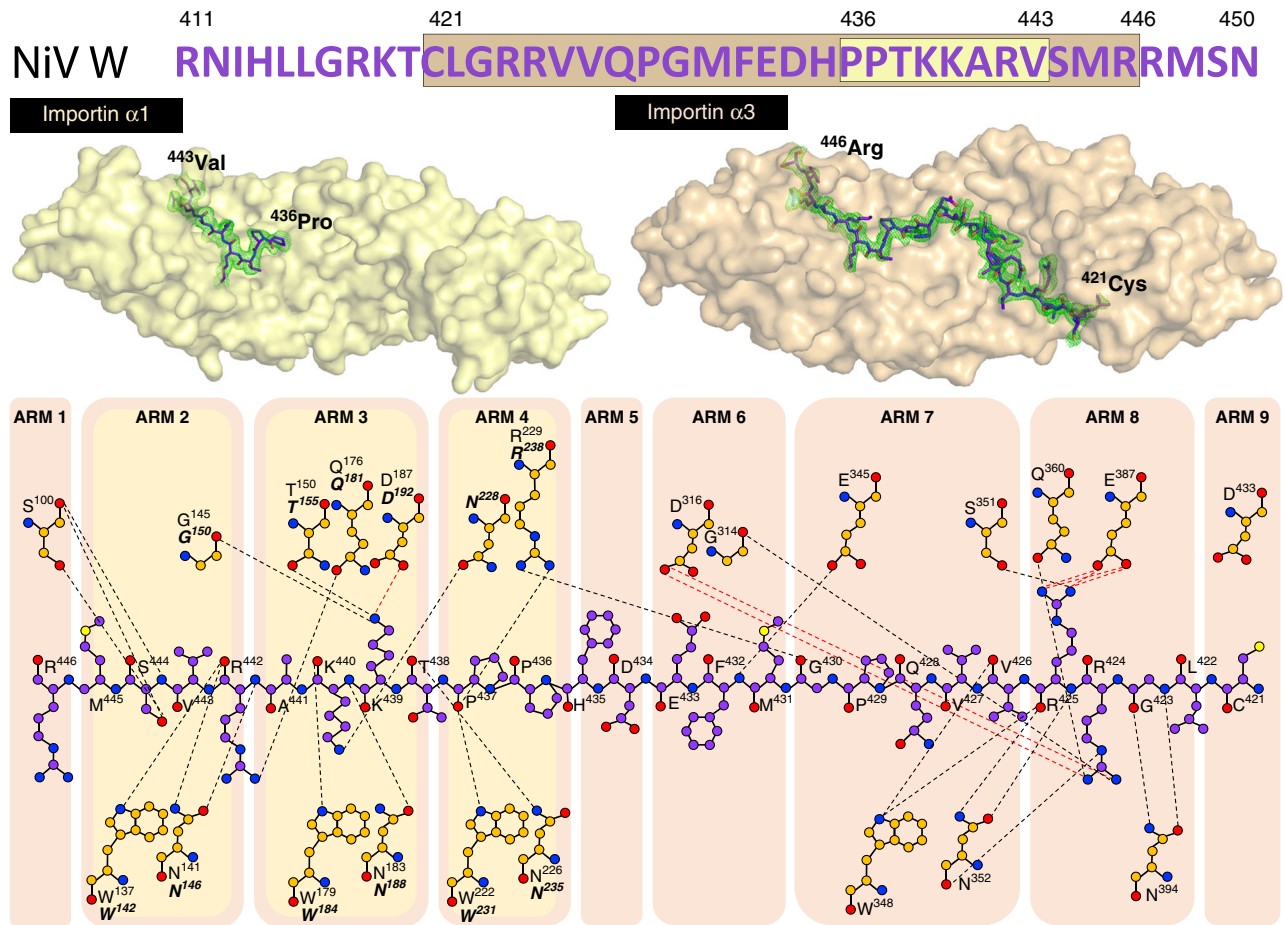

**Fig. 3** Structural comparisons and basis for high-affinity interaction of NiV W with importin α3. Importin α1:NiV W and importin α3:NiV W structures were solved to 2.1 Å and 2.3 Å resolution, respectively. The importin α adaptors are shown in cartoon and surface representation (importin α1: bright yellow, importin α3: light orange) and NiV W NLS in stick representation (magenta: carbon atoms, red: oxygen atoms, blue: nitrogen atoms) with associated simulated annealing Fo-Fc omit maps of the NLSs contoured to 3σ in green. Schematics of the binding interface and specific interactions are shown below (magenta: carbon atoms, red: oxygen atoms, blue: nitrogen atoms), with hydrogen bond and salt bridge interactions depicted by dash lines (black and red, respectively), and the partner interactions for importin α1 (bold italics) and importin α3 coloured as orange: carbon atoms, red: oxygen atoms, blue: nitrogen atoms. The location of these interactions in the ARM repeats highlight the greater interaction interface of the NiV W NLS for the importin α3 adaptor

conformation by superimposing the HeV W NLS from importin α3 onto the importin α1 structure. Although almost all other ARM repeat interactions remained the same, ARM repeats 7 and 8 of importin α1 clashed with the HeV W C terminus (Fig. 5b), indicating that the positioning of these ARM repeats in importin α3 allows for favourable interactions that are not possible with importin α1.

As these ARM repeats appeared to be important for isoform specificity, we tested whether these regions were flexible and able to adapt to accommodate different NLSs. We superimposed importin α1 structures bound to a variety of NLSs including monopartite and bipartite, NLSs within domains including CAP80, and BiMax, and found that in all structures the importin α1 C-terminal ARM repeats 7 and 8 were in the same position (Fig. 5c). Similarly, superposition of a variety of importin α3 structures confirmed these regions are highly similar in all structures and allow for favourable binding (Fig. 5d). This supports the notion that the positioning of these ARM regions, highly similar across a wide range of structures, is conducive to high affinity binding of W in importin α3, but not importin α1.

**Cargo-free importin α1 and α3 retain position of ARMs 7–8.** To extend on the structural repertoire of importin α structures, we tested whether these ARM repeats are similar in the structures of importin α isoforms that do not contain bound cargo. Importin α1 and α3, crystallised in $P2_12_12_1$ and $P12_11$ space groups, respectively, both contained one chain in the ASU. This is the first report of these adaptors being crystallised without cargo bound in the NLS-binding pockets, although a dimer of importin α1 has been reported previously[33]. Comparison of the cargo-free and W NLS-bound forms of importin α1 showed that the structures were remarkably similar with an r.m.s.d of 0.180 Å across 408 residues, and importantly, the positioning of ARM repeats 7 and 8 remain unchanged (Fig. 6a). Superposition of importin α3 in the cargo free and bound forms showed that the N-terminal ARM repeats 1–4 are repositioned by up to 5 Å (Fig. 6b), consistent with previous findings that a more flexible hinge region is present in importin α3[15]. However, the position of ARM repeats 7 and 8 was retained in both the cargo free and bound forms of importin α3. These results suggest that ARMs 7 and 8 do not undergo conformational changes to accommodate binding of the W NLS in either importin α1 or α3. Significantly, although the flexible hinge

**Table 1 Data collection and refinement statistics for importin α1 structures**

| | Impα1[a] | NiV W Impα1[a] | HeV W Impα1[a] |
|---|---|---|---|
| *Data Collection* | | | |
| Wavelength (Å) | 0.9537 | 0.9537 | 0.9537 |
| Space group | P 2$_1$ 2$_1$ 2$_1$ | P 2$_1$ 2$_1$ 2$_1$ | P 2$_1$ 2$_1$ 2$_1$ |
| Cell dimensions | | | |
| *a, b, c* (Å) | 78.5, 90.0, 100.7 | 77.9, 88.8, 97.6 | 79.0, 89.3, 100.4 |
| *α, β, γ* (°) | 90, 90, 90 | 90, 90, 90 | 90, 90, 90 |
| Resolution (Å) | 24.54–2.50 | 29.61–2.10 | 19.76–2.20 |
| | (2.60–2.50) | (2.16–2.10) | (2.28–2.20) |
| $R_{pim}$ | 0.061 (0.365) | 0.033 (0.253) | 0.029 (0.147) |
| Mean $I/\sigma$ (I) | 10.2 (2.9) | 11.7 (2.4) | 18.5 (5.4) |
| CC$_{1/2}$ | 0.989 (0.751) | 0.998 (0.814) | 0.999 (0.928) |
| Total reflections | 154,003 (17460) | 136,253 (11369) | 208,967 (18156) |
| Unique reflections | 24,805 (2822) | 38,272 (3193) | 36,721 (3139) |
| Completeness (%) | 98.6 (100) | 96 (97.8) | 99.8 (100) |
| Redundancy | 6.2 (6.2) | 3.6 (3.6) | 5.7 (5.8) |
| Wilson *B*-factor | 26.4 | 31.8 | 22.6 |
| *Refinement* | | | |
| Resolution (Å) | 24.54–2.50 | 28.79–2.10 | 19.76–2.20 |
| Reflections used in | | | |
| Refinement | 24,772 (2478) | 38,216 (3843) | 36,662 (3611) |
| $R_{free}$ | 1262 (133) | 1864 (172) | 1861 (176) |
| $R_{work}$ | 0.1910 (0.2227) | 0.1854 (0.2690) | 0.1915 (0.2257) |
| $R_{free}$ | 0.2129 (0.2572) | 0.2032 (0.3062) | 0.2121 (0.2774) |
| Number of non-hydrogen atoms | 3361 | 3555 | 3431 |
| Macromolecules | 3244 | 3306 | 3306 |
| Protein residues | 426 | 434 | 434 |
| *B* factors | 42.49 | 51.14 | 38.89 |
| Protein | 42.66 | 51.31 | 38.98 |
| Water | 37.95 | 48.89 | 36.67 |
| R.M.S. deviations | | | |
| Bond lengths (Å) | 0.004 | 0.005 | 0.003 |
| Bond angles (°) | 0.59 | 0.73 | 0.62 |
| Ramachandran plot (%) | | | |
| Favoured | 99 | 99 | 98 |
| Allowed | 1.4 | 1.4 | 1.9 |
| Outliers | 0 | 0 | 0 |
| Rotamer outliers (%) | 1.1 | 1.1 | 0.27 |
| Clash score | 0.46 | 0.90 | 1.19 |

[a]Values in parentheses are for highest resolution shell

region of importin α3 is unlikely to play a role in cargo containing an NLS in the C terminus of the protein (as the NLS ends before reaching the hinge region), the positioning of ARM repeats 1–3 could play a role in the binding of proteins harbouring an NLS at the N terminus, such as RCC1 (see Discussion).

**Chimera of importin α1 ARMs 1-5:α3 ARMs 6-10 retains binding**. To test our hypothesis that the positioning of ARM repeats 7 and 8, located in the C terminus of importin α3, mediate the isoform specificity for W protein, we examined whether chimeras comprised of importin α1 and importin α3 could mediate binding to W proteins (Fig. 7a). We found that a chimera comprised of the N terminus of importin α1 (IBB domain and ARMs 1–5) and the C terminus of importin α3 (ARMs 6–10) (importin α1$^{ARM1-5}$:α3$^{ARM6-10}$) bound W, whereas the reverse chimera comprised of importin α3 IBB ARMs 1–5 and importin α1 ARMs 6–10 (importin α3$^{ARM1-5}$:α1$^{ARM6-10}$) did not pull-down W (Fig. 7b). These results support the structural data that the C-terminal domain of importin α3 mediates isoform specificity for the W protein. Moreover, the importin α1$^{ARM1-5}$: α3$^{ARM6-10}$ bound with 8.3 nM affinity to HeV W, and 4.7 nM

affinity to NiV W, comparable to the binding affinity of importin α3 (Supplementary Fig. 5). To further test the involvement of ARMs 7 and 8 in mediating specificity, we designed an importin α3 mutant based on the structural data to disrupt interactions in ARMs 7/8 (N348 A/N352A/E387A/N394A), and found a marked reduction in binding to both HeV and NiV W proteins (Fig. 7c). Confirming their functionality, both the importin α1$^{ARM1-5}$: α3$^{ARM6-10}$ chimera and importin α3 ARMs 7/8 mutant retained the ability to bind to the SV40Tag classical NLS (Supplementary Fig. 6). The importin α3$^{ARM1-5}$:α1$^{ARM6-10}$ formed inclusion bodies during overexpression and could not be purified. These results, together with the structural data, suggest that the positioning of the C-terminal ARM repeats 7 and 8 are important for mediating isoform specificity of henipavirus W proteins.

## Discussion

In this study, we present high-resolution structures of importin α isoform adaptors bound to the NLS regions of HeV W and NiV W, providing insights into the molecular basis of importin α isoform specificity. We found that HeV W and NiV W proteins bind importin α3 preferentially, and that these NLS regions

**Table 2 Data collection and refinement statistics for importin α3 structures**

| | Impα3[a] | NiV W Impα3[a] | HeV W Impα3[a] crystal form 1 | HeV W Impα3[a] crystal form 2 | HeV W Impα3[a] crystal form 3 |
|---|---|---|---|---|---|
| *Data collection* | | | | | |
| Wavelength (Å) | 0.9537 | 0.9537 | 0.9537 | 0.9537 | 0.9537 |
| Space group | P 1 2$_1$ 1 | P 1 2$_1$ 1 | P 1 2$_1$ 1 | P 1 2$_1$ 1 | P 2$_1$ 2$_1$ 2$_1$ |
| Cell dimensions | | | | | |
| $a, b, c$ (Å) | 47.1, 53.2, 91.3 | 47.7, 65.6, 73.9 | 47.4, 65.7, 74.2 | 48.1, 60.0, 89.9 | 48.2, 59.0, 169.3 |
| $\alpha, \beta, \gamma$ (°) | 90, 92.87, 90 | 90, 99.79, 90 | 90, 99.76, 90 | 90, 97.85, 90 | 90, 90, 90 |
| Resolution (Å) | 19.81–2.30 (2.38–2.30)[a] | 38.21–2.30 (2.38–2.30) | 19.84–1.6 (1.63–1.60) | 19.95–2.20 (2.27–2.20) | 29.49–2.30 (2.38–2.30) |
| Rpim | 0.086 (0.304) | 0.087 (0.274) | 0.061 (0.174) | 0.081 (0.369) | 0.032 (0.077) |
| Mean $I/\sigma$ (I) | 11.3 (1.8) | 8.6 (3.5) | 7.1 (3.3) | 9.6 (3.7) | 17.1 (8.8) |
| CC$_{1/2}$ | 0.977 (0.785) | 0.781 (0.817) | 0.989 (0.899) | 0.974 (0.446) | 0.997 (0.973) |
| Total reflections | 105,917 (11,952) | 46,617 (4355) | 254,011 (12,863) | 90,676 (7765) | 286,394 (27,948) |
| Unique reflections | 19,290 (1942) | 19,881 (1953) | 58,984 (2917) | 25,833 (2232) | 22,252 (2139) |
| Completeness (%) | 95 (98.2) | 98.9 (99.2) | 99.4 (99.6) | 99.6 (99.8) | 100 (100) |
| Redundancy | 5.5 (6.2) | 2.3 (2.2) | 4.3 (4.4) | 3.5 (3.5) | 12.9 (13.1) |
| Wilson B-factor | 37.1 | 20.4 | 10.7 | 15.5 | 15.1 |
| *Refinement* | | | | | |
| Resolution (Å) | 19.48–2.30 | 36.75–2.30 | 19.58–1.60 | 19.95–2.20 | 29.36–2.30 |
| Reflections used in | | | | | |
| Refinement | 19,147 (1978) | 19,788 (1980) | 58,886 (5838) | 25,804 (2546) | 22,192 (2161) |
| $R_{free}$ | 937 (91) | 989 (94) | 2894 (275) | 1227 (115) | 1119 (123) |
| $R_{work}$ | 0.2500 (0.2662) | 0.1991 (0.2177) | 0.1711 (0.1978) | 0.1859 (0.2606) | 0.1771 (0.1929) |
| $R_{free}$ | 0.2684 (0.3212) | 0.2223 (0.2917) | 0.1962 (0.2421) | 0.2148 (0.2946) | 0.2182 (0.2578) |
| Number of non-hydrogen atoms | 3210 | 3592 | 3738 | 3678 | 3624 |
| Macromolecules | 3206 | 3400 | 3400 | 3419 | 3419 |
| Protein residues | 415 | 440 | 440 | 442 | 442 |
| B factors | 55.53 | 30.56 | 18.57 | 30.72 | 27.54 |
| Protein | 55.56 | 30.59 | 17.77 | 30.47 | 27.46 |
| Water | 35.29 | 30.04 | 26.57 | 33.99 | 28.91 |
| R.M.S. deviations | | | | | |
| Bond lengths (Å) | 0.005 | 0.006 | 0.015 | 0.003 | 0.006 |
| Bond angles (°) | 1.03 | 0.75 | 1.21 | 0.54 | 0.71 |
| Ramachandran plot (%) | | | | | |
| Favoured | 99 | 98 | 99 | 98 | 98 |
| Allowed | 1.2 | 2.1 | 1.4 | 2.3 | 2.1 |
| Outliers | 0 | 0 | 0 | 0 | 0 |
| Rotamer outliers (%) | 0.83 | 0.26 | 0.52 | 0.52 | 0.52 |
| Clash score | 3.57 | 1.03 | 1.76 | 0.87 | 1.89 |

[a]Values in parentheses are for highest resolution shell

interact with higher affinity over other importin α subfamilies. Using structural approaches, we identified key features that account for adaptor specificity. We found that the isoform specificity was localised to the C-terminal ARM repeats 7 and 8, and that the positioning of these domains was an important determinant for mediating specificity.

Although numerous studies have reported specificity of nuclear adaptors for a wide range of cargo[19,34–37] and associated function of these interactions in many diseases[36], the mechanism(s) behind NLS adaptor specificity has remained elusive. Recently, one study described the specificity of the RCC1 factor for importin α3, highlighting that additional residues outside the NLS were important for maintaining specificity[15]. A comparison between the mechanism presented in the RCC1 study and our study, highlights a number of important differences. The study by Sankhala et al. [15] indicated that the NLS of RCC1 binds in the major binding site of ARMS 2–4 of both importin α1 and α3, with additional interactions occurring at the N terminus of importin α3 ARM repeats 1–4 and the β-propeller region of RCC1,

mediated by flexibility and rotation of importin α3 in ARM repeats 1–2. This mechanism of isoform specificity is well suited to cargo containing NLSs at the N terminus, and our structural data of the unbound and NLS-bound forms of importin α3 supports this model. This mechanism is distinct, however, from that governing specificity for cargo such as the W proteins in this study, where the NLSs are located at the C terminus because the protein interface would not extend past the N terminus of ARM repeats 2–4. Thus, it may be possible that isoform specificity of cargo containing N-terminal NLSs may reside through differential interaction of the N-terminal ARM repeats, whereas cargo bearing C-terminal NLSs may show specificity through differential interaction with the C terminus of importins, as demonstrated in this study. Although further work will be needed to establish how extensive these rules are, it is likely that different mechanisms may govern isoform specificity, dependent upon the position of the NLS within the cargo.

The findings of our study are distinct from mechanisms that have been previously hypothesised. The IBB domain has been

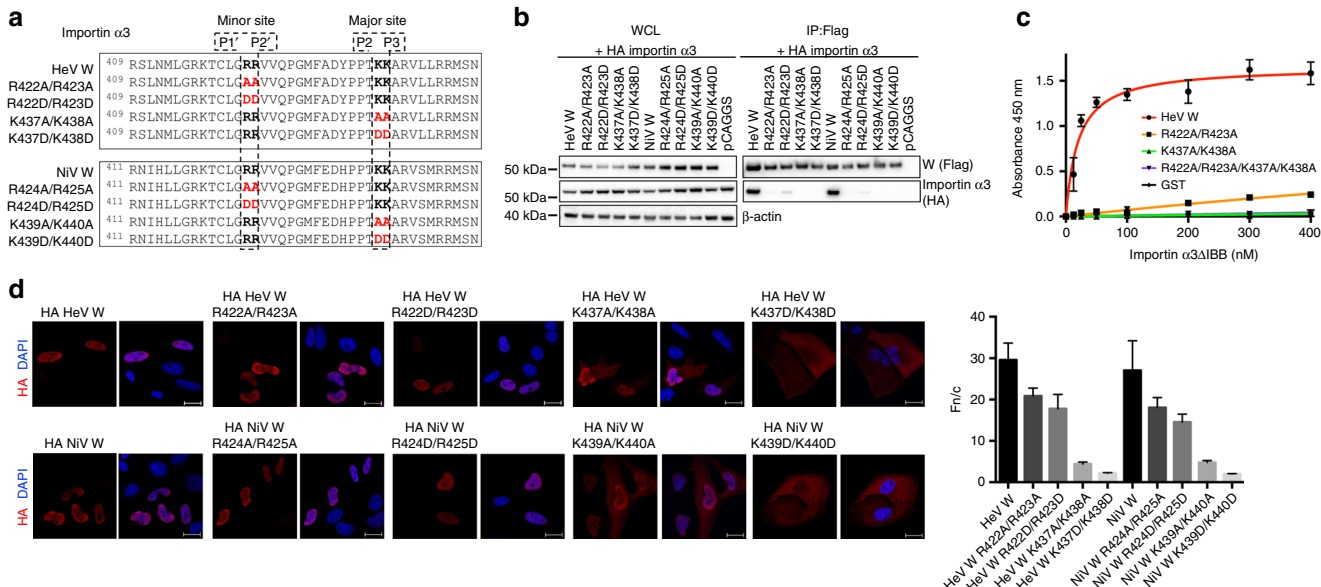

**Fig. 4** Both importin α3 major and minor sites are important for the HeV NiV W protein interactions. **a** Schematic of the mutations made in the W protein to test the binding contributions at the major and minor binding sites for importin α3. **b** Co-immunoprecipitation assay performed with Flag antibody on lysates of HEK293T cells expressing HA-tagged importin α3 and Flag-tagged HeV W, NiV W, and mutant W constructs as indicated. Western blots were performed for HA and Flag. WCL, whole cell lysate; IP, immunoprecipitation. pCAGGS denotes empty vector control. **c** HeV W mutations were tested against importin α3 using the ELISA assay. GST-HeV W and mutants were coated on 96-well plates, and binding of 6xHis-tagged importin α3 assessed using an anti-6xHis HRP antibody. Error bars show the S.E.M for three replicates. **d** Hela cells were transfected with indicated HA-tagged HeV and NiV W plasmids. Twenty-four hours post transfection, cells were fixed and stained with anti-HA antibody and Alexa Fluor 594 to visualise protein localisation. Image is representative. Scale bar depicted is 20 μm. The ratio of nuclear to cytoplasmic fluorescence (Fn/c) was determined for 50 cells for each construct, error bars represent the SEM

shown to auto-inhibit differentially across isoforms[19]. However, our results identify clear differences in the structural positioning of the C-terminal ARM domains 7–8 of importin α1 and α3, accompanied with a significant increase in the binding interface at this region in importin α3. In addition, the chimeric protein importin α1$^{ARM1-5}$:α3$^{ARM6-10}$ binds to W, whereas importin α3$^{ARM1-5}$:α1$^{ARM6-10}$ does not interact, confirming that the C-terminal domain of the importin α is the differentiating factor, rather than the N terminus. The importance of the C-terminal ARM repeats has been demonstrated for importin α5 binding of nonclassical NLS cargo, involving a distinct mechanism and ARM repeat 10[38]. It has also been hypothesised that that flexibility of importin α3 may mediate increased specificity and/or affinity, and indeed, this was described as a contributing factor to the binding of RCC1. However, this is unlikely to have a role in the specificity of importin α3 for the W protein, because the difference in binding interactions occur outside of the hinge region. As well, if the hinge region was critical for mediating specificity, the α3$^{ARM1-5}$:α1$^{ARM6-10}$ chimera would have interacted with W if the hinge region was the critical region for mediating specificity. In addition, we show through structural comparisons between apo- and W bound-importin α3, that structural changes in the N terminus were not associated with direct binding interactions in this region. Importantly, these results do not discount the previous work of RCC1 binding, but rather, complement and extend it by highlighting that different mechanisms are likely to exist in mediating specificity.

Our study defines a basis to explain the importin α specificity of henipavirus HeV W and NiV W virulence factors for importins, demonstrating that differences in the ARM repeats in the C terminus of importin α3 mediate specificity. Nuclear transport of cargo in an isoform and tissue-specific manner is critical for health and disease. Subtype switching of importin α1 to α5 in

embryonic stem cells results in the initiation of neural differentiation[39], and the maintenance and lineage determination of embryonic stem cells[40]. Differential expression of importin α3 has been demonstrated to occur following rabies infection, with overexpression observed only in paralytic rabies, suggesting a possible prognostic marker[41]. The inhibition of importin α3 has also been shown to attenuate prostate cancer mestastasis[42]. The expanded rules for isoform specificity outlined here will provide important insights into our understanding of nuclear transport adaptor cargo specificity and their function in cellular processes.

## Methods

**Plasmids for recombinant protein expression.** The C-terminal domain of HeV (residues 409–448; UniProtKB P0C1C6) and NiV (residues 411–450; UniProtKB P0C1C7) with an N-terminal TEV cleavage site (ENLYFQS) were codon optimised for expression in *Escherichia coli* and synthesised (Genscript, Piscataway, NJ). These constructs were cloned into pGEX4T-1 vector at *Bam*HI and *Eco*RI sites. Importin α3 and α7 lacking the auto-inhibitory IBB domain (residues 64–521 and 73–536) were codon optimised for *E. coli* expression, synthesised and cloned into pET15b vector at *Nde*I and *Eco*RI sites. The importin α chimeras lacking the auto-inhibitory IBB domain (for recombinant expression) were comprised of α1aa71–279:α3aa271–521 and α3aa64–270:α1aa280–529, with numbering according to UniProtKB P52292 and UniProtKB O00629 for importin α1 and importin α3, respectively. Importin α1ΔIBB, encoding residues 71–529, was cloned in the pET30 vector as described previously[43,44].

**Recombinant expression and purification.** Plasmids were transformed into BL21 (DE3) pLysS cells and expressed using the Studier auto-induction method[45]. In brief, starter cultures were inoculated into 2 L baffled flasks containing 500 mL of expression media consisting of 1% (w/v) tryptone, 0.5% (w/v) yeast extract, 0.5% glycerol, 0.05% glucose, 0.2% (w/v) α-lactose, 0.025 M $NH_4SO_4$, 0.05 M $KH_2PO_4$, 0.05 M $Na_2HPO_4$, 1 mM magnesium chloride and either 100 μg/mL ampicillin, or 50 μg/mL kanamycin. Cells were harvested via centrifugation at 6500×*g* and 18 °C for 30 min and resuspended in phosphate buffer (PB) (20 mM imidazole, 300 mM NaCl, 50 mM phosphate pH 8.0) or Tris-buffered saline (TBS) (Tris pH 8.0, 125 mM NaCl) buffer with complete ethylenediaminetetraacetic acid (EDTA)-free

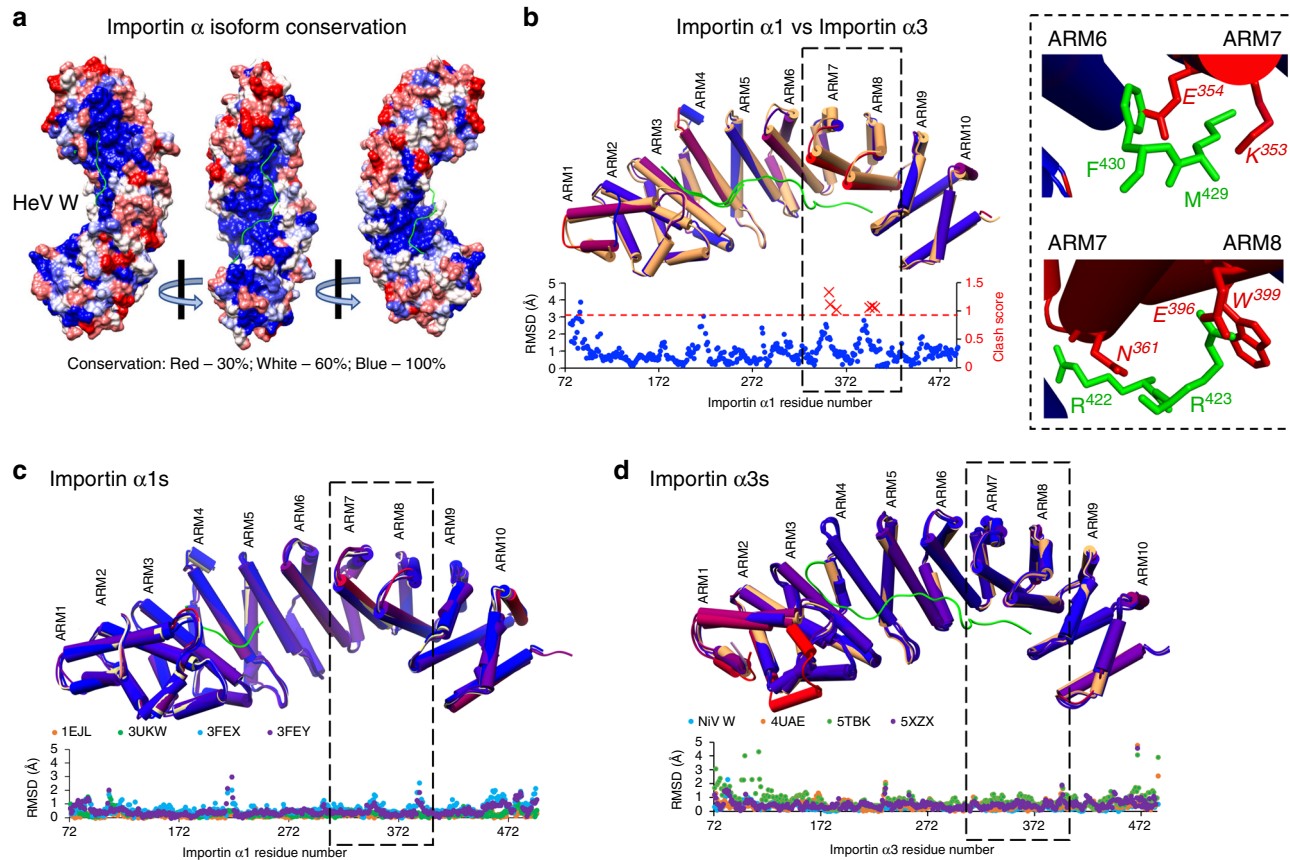

**Fig. 5** Structural basis for specificity of Henipavirus W binding to the importin α3. **a** Importin α3:HeV W structure coloured by conservation of amino acids across all importin α isoforms highlights a 100% conservation in the binding interface. The HeV W NLS backbone (coloured green) is shown in complex with importin α3, with sequence colour rendering red at 30%, white 60%, and blue 100% sequence identity. Figure created in UCSF Chimera using importin α alignments from Clustal Omega. **b** To identify structural differences, the structures of importin α1:HeV W and importin α3:HeV W were superimposed in UCSF Chimera using MatchEnsemble, and r.m.s.d. plots generated using MatchAlign, and MatchAssess functions. α-helices shown as cylinders; importin α3 coloured orange throughout, and colour conservation rendering for importin α1 set for blue, < 2.5 Å variance, and red for variances > 2.5 Å. Molprobity was used to analyse clash data of importin α3:HeV W NLS superimposed to importin α1, all clashes > 0.8 shown as red crosses. Differences in clash score between importin α1 and α3 are localised to ARMs 7 and 8. Detailed view of clashes are presented in the right box, highlighting residues clashing with the W NLS owing to the difference in positioning of ARMs 7 and 8 in importin α1. All clashing residues are positioned on the α-helices of the ARMs. **c** Structural comparisons of importin α1 bound to a range of cargo was examined by superimposing the α1:HeV W NLS structure determined in this study (reference molecule, coloured yellow), with importin α1 bound to a monopartite SV40T NLS[12] (1EJL), bipartite Bimax NLS[59,60] (3UKW), and two domain bound structures of CAP80[61] (3FEX, 3FEY) coloured according to r.m.s.d as described in **b**. Positioning of the α-helices in ARMs 7 and 8 are relatively static across all structures. **d** Structural comparisons of importin α3 bound to a range of cargo was examined by superimposing the α3:HeV W NLS structure determined in this study (reference molecule coloured orange) with the monopartite RanBP3[62] (5ZXX), the NiV W (this study), and domains of PB2[19] (4UAE) and RCC1[15] (5TBK) coloured according to r.m.s.d as described in **b**. Positioning of α-helices in ARMs 7 and 8 are also relatively static across all structures

protease inhibitor. Cells were lysed using two freeze–thaw cycles and addition of 20 mg lysozyme and 0.5 mg DNase.

Affinity purifications of 6xHis-tagged importin α were performed by injecting clarified cell lysate onto a GE HisTrap 5 mL column using PB, washing the column with 15 column volumes and then eluting over 5 column volumes using a gradient elution with high imidazole PB (500 mM imidazole, 300 mM NaCl, 50 mM phosphate pH 8.0). Affinity purifications of GST-tagged proteins were performed on a GST Trap 5 mL column using TBS, and elution buffer containing 10 mM glutathione. All size exclusion purifications were performed on a Superdex 200 pg 26/600 column using TBS pH 8.0, and eluted proteins were pooled and concentrated using 10 kDa MW centrifuge filters. Complex formation was performed as described previously[46].

**ELISA**. The method was based on previously published microtiter plate assays[47]. In brief, 96-well clear plates were coated with GST-NLS fusion proteins using bicarbonate/carbonate buffer pH 9.6 for 2 h at room temperature. The plates were washed using TBS containing 0.05% v/v Tween20 (TBST). Blocking was achieved using 5% (w/v) skim milk in TBST for 2 h at room temperature. Wells were washed three times in TBST buffer and incubated with decreasing concentrations of 6xHis-

tagged importin αΔIBB (400, 300, 200, 100, 50, 25, 12.5 and 0 nM) diluted in TBS for 2 h. The plate was washed three times, blocked for 2 h, washed a further three times and then incubated for 2 h with 100 µL a 1/5000 dilution of anti-6xHis 4HRP conjugated rabbit polyclonal antibody (Abcam ab1187). The plate was washed a further three times before addition of 100 µL of TMB substrate (Sigma T4444). The colorimetric reaction proceeded for 20 min before being stopped with 100 µL of 2 M $H_2SO_4$, and the absorbance measured at 450 nm using an Epoch microplate spectrophotometer (Biotek). One-site specific binding analysis using least squares fit was performed using GraphPad Prism version 7.00 for Mac, GraphPad Software, La Jolla California USA, www.graphpad.com.

**Microscale thermophoresis**. Affinity measurements using MST were carried out employing a Monolith NT.115 instrument (NanoTemper Technologies)[48]. Purified importin α1, α3 and α7 in 20 mM 4-(2-hydroxyethyl)-1-piperazineethanesulfonic acid (HEPES), 150 mM NaCl, pH 8.0 were labelled using the NHS RED Nano-Temper labelling kit according to the manufacturer's instructions. For the assay, 5 µL of labelled protein was mixed with 10 µL of the unlabelled NiV W at various concentrations and 5 µL of 0.05% (w/v) Tween20. All experiments were incubated for 10 min before applying samples to Monolith NT Standard Treated Capillaries

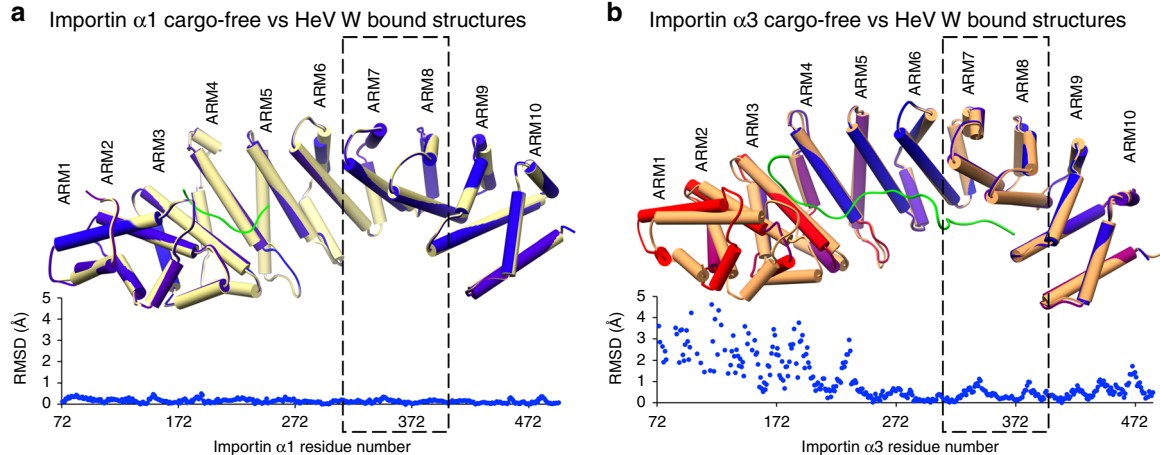

**Fig. 6** Structures of cargo-free importin α1 and α3 confirm ARMs 7 and 8 positioning. **a** Importin α1 cargo-free structure and structural superposition with importin α1:HeV W complex (reference molecule, coloured yellow) and associated r.m.s.d plot. The positioning of the α-helices in ARM repeats 7 and 8 are static across all structures. The importin α1 is coloured by r.m.s.d rendering where blue < 2.5 Å variance, and red for variances above 2.5 Å. **b** Importin α3 cargo-free structure, together with the associated structural superposition with importin α3:HeV W complex, and r.m.s.d plot. Importin α structures were aligned in UCSF Chimera using MatchEnsemble, and r.m.s.d. plots were generated using MatchAlign, and MatchAssess functions. Colouring as per **a**, but with importin α3:HeV W in orange

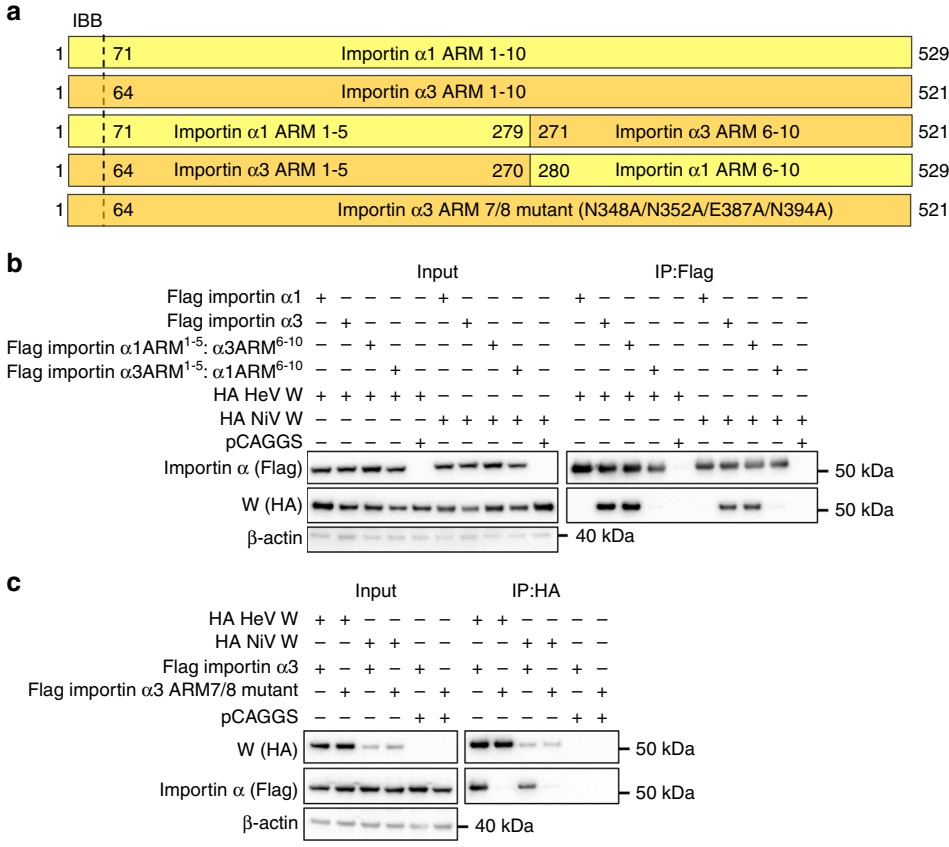

**Fig. 7** Confirming the structural hypothesis through importin α chimeras. **a** Schematic representation of importin α1, α3, and chimeric importin α1[ARM1-5]: α3[ARM6-10] and importin α3[ARM1-5]:α1[ARM6-10] constructs. **b** Co-immunoprecipitation assay performed with Flag antibody on lysates of HEK293T cells expressing Flag-tagged importin α1, α3, importin α1[ARM1-5]:α3[ARM6-10] and importin α3[ARM1-5]:α1[ARM6-10], and HA-tagged HeV W and NiV W, as indicated. Western blots were performed for HA and Flag. WCL, whole cell lysate; IP, immunoprecipitation. pCAGGS denotes empty vector control. **c** Co-immunoprecipitation assay performed with HA antibody on lysates of HEK293T cells expressing Flag-tagged importin α1, α3, importin α1[ARM1-5]:α3[ARM6-10] and importin α3[ARM1-5]:α1[ARM6-10], and HA-tagged HeV W and NiV W, as indicated. Western blots were performed for HA and Flag. WCL, whole-cell lysate; IP, immunoprecipitation. pCAGGS denotes empty vector control

(NanoTemper Technologies). Thermophoresis was measured at 25 °C with laser off/on/off times of 5 s/30 s/5 s. Experiments were conducted at 20% light-emitting diode power and 20–40% MST infra-red laser power. Data from three independently performed experiments were fitted to the single binding model via the NT. Analysis software version 1.5.41 (NanoTemper Technologies) using the signal from Thermophoresis + T-Jump.

**Crystallisation**. All crystals were obtained using the hanging drop vapour diffusion method over a 300 µL reservoir solution. Importin α1ΔIBB was crystallised with 1 M NH$_4$SO$_4$, 0.01 M DTT, 0.1 M sodium HEPES pH 7.0. The importin α1ΔIBB:NiV W and importin α1ΔIBB:HeV W complexes were crystallised in 0.01 M DTT, 0.7 M sodium citrate and 0.1 M sodium HEPES pH 7.0, with rod-shaped crystals forming within 2–3 days. Importin α3ΔIBB was crystallised with 0.2 M sodium nitrate, 0.1 M Bis-Tris propane pH 6.5 and 25% (w/v) PEG 3350. Plate-like crystals formed within 2–3 weeks. The importin α3ΔIBB:NiV W complex crystallised in 0.2 M lithium nitrate, 20% (w/v) PEG 3350 conditions with a rod morphology that diffracted to 2.3 Å. The importin α3ΔIBB:HeV W complex crystallised in three forms. Crystal form 1 was obtained using 0.1 M sodium citrate pH 5.0 and 20% (w/v) PEG 2000, crystal form 2 was obtained using 0.1 M sodium HEPES pH 7.5, 25% (w/v) PEG 200 MME and crystal form 3 obtained using 0.2 M sodium chloride, 0.1 M MES pH 6.5, 10% (w/v) PEG 4000 conditions. Diffraction of the importin α3ΔIBB:HeV W complex crystal forms ranged from 1.6–2.3 Å.

**Data collection and processing**. X-ray diffraction data were collected at the Australian Synchrotron on the MX1[49] and MX2[50] macromolecular beam lines using an ASDC Quantum 210r, ASDC Quantum 315r detector and Eiger 16 M detector, respectively. Data reduction and integration was performed using iMosflm[51] for data collected using ADSC 210r and ADSC 315r detectors, whereas reduction and integration of data collected on Eiger 16 M was performed using XDS[52]. Merging, space group assignment, scaling and selection of 5% reflections for R$_{free}$ calculations was done using Aimless[53,54] and the CCP4 suite[55]. Phasing was performed using molecular replacement in Phaser MR[56], with PDBID 5FC8 used as a search model for importin α1 cargo-free, importin α1:HeV W complex and importin α1:NiV W. The importin α3:HeV W complex was phased using 4UAE as a search model, from which the solution was then used as a search model for the importin α3:NiV W complex. The cargo-free importin α3 structure was phased by placing the N terminus and C terminus domains separately. Models were refined using iterative cycles of manual real space coot[57] and maximum likelihood phenix refine[58].

**Cell culture and plasmids**. HEK293T (ATCC—CRL-3216) and Hela cells (ATCC —CCL-2) were maintained in Dulbecco's Modified Eagle Medium, supplemented with 10% fetal bovine serum and cultured at 37 °C and 5% CO$_2$.

The sequence for HeV W (NCBI: JN255804.1) was synthesised (Genscript, Piscataway, NJ) and cloned with N-terminal Flag- and HA-tags into the mammalian expression plasmid pCAGGS. pCAGGS HA NiV W was previously described[20]. NiV W was subcloned with a Flag-tag into pCAGGS. Overlapping PCR was used to clone HeV W R422A/R423A, HeV W R422D/R423D, HeV W K437A/K438A, HeV W K437D/K438D, NiV W R424A/R425A, NiV W R424D/ R425D, NiV W K439A/K440A, NiV W K439D/K440D and importin α3 ARMs 7/8 mutant (N348 A/N352A/E387A/N394A), which were then cloned with an HA- or Flag-tag into pCAGGS. pCAGGS Flag importin α1, α3, α4, α5, α6 and α7 were previously described[20,36]. Importin α3 was subcloned with an HA-tag into pCAGGS. Overlapping PCR was used to generate the chimeric importin α1$^{ARM1-5}$: α3$^{ARM6-10}$ (residues 1–279 of importin α1 and residues 271–521 of importin α3) and importin α3$^{ARM1-5}$:α1$^{ARM6-10}$ (residues 1–270 of importin α3 and residues 280–529 of importin α1), which were cloned with Flag tags into pCAGGS. For additional information regarding primers used in this study, please see Supplementary Table 6.

**Co-immunoprecipitation assays**. HEK293T cells ($1 \times 10^6$) were transfected with the indicated plasmids using Lipofectamine 2000 (Thermo Fisher Scientific, MA) and at 24 h post transfection, cells were lysed in NP-40 lysis buffer (50 mM Tris pH 7.5, 280 mM NaCl, 0.5% Nonidet P-40, 0.2 mM EDTA, 2 mM ethylene glycol-bis(β-aminoethyl ether)-N,N,N',N'-tetraacetic acid, 10% glycerol, protease inhibitor (complete; Roche, Indianapolis, IN)). Anti-FLAG M2 magnetic beads or EZview Red anti-HA Agarose affinity gel (Sigma-Aldrich, St. Louis, MO) were incubated as indicated with lysates for 1 h at 4 °C, washed five times in NP-40 lysis buffer, and eluted using 3X FLAG or HA peptide (Sigma-Aldrich, St. Louis, MO) at 4 °C for 30 min. Whole cell lysates and co-precipitation samples were analysed by western blot.

**Immunofluorescence**. Hela cells ($3 \times 10^4$) grown on glass coverslips were transfected with indicated HeV and NiV W plasmids (300 ng) using Lipofectamine 2000 (Thermo Fisher Scientific, MA). At 24 h post transfection, cells were fixed using 4% paraformaldehyde and permeabilised using 0.1% Triton X-100. Cells were stained using mouse anti-HA (H3663, Sigma-Aldrich, MO) (dilution 1:500) and secondary antibody conjugated to Alexa Fluor 594 (A-11032, Thermo Fisher Scientific, MA) (dilution 1:2000). Images were taken using a Zeiss LSM 800 confocal microscope at × 64. To determine the ratio of nuclear to cytoplasmic fluorescence signal (Fn/c),

Hela cells ($1 \times 10^4$) were plated in a 96-well plate (black, clear bottom, Corning) and transfected with the indicated HeV and NiV W plasmids (300 ng) using Lipofectamine 2000. At 24 h post transfection, cells were fixed and permeabilised as above and stained with anti-HA (H6908, Sigma-Aldrich, MO) (dilution 1:500) and Alexa Fluor 488 (A32731, Thermo Fisher Scientific, MA) (dilution 1:2000). Images were taken at × 10 using a BioTek Cytation 5 Cell Imaging Multi-Mode reader and were analysed using Gen5 Image Prime software to determine Fn/c, using the calculation Fn/c = (Fn-background)/(Fc-background), where Fn is nuclear fluorescence and Fc is cytoplasmic fluorescence. Fn/c was determined for 50 cells in each condition; error bars indicate the standard error of the mean (SEM).

**Antibodies**. Polyclonal rabbit anti-Flag (F7425), polyclonal rabbit anti-HA (H6908), and monoclonal mouse anti-HA (H3663) were purchased from Sigma-Aldrich (St. Louis, MO). Monoclonal rabbit β-actin antibody was purchased from Cell Signalling (4967 S) (Danvers, MA).

**Western blots**. Lysates were run on 10% Bis-Tris Plus polyacrylamide gels (Thermo Fisher Scientific, MA) and transferred to polyvinylidene difluoride membrane (Bio-Rad, Hercules, CA). Membranes were probed with the indicated antibodies and were developed using Western Lightning Plus ECL (Perkin Elmer, Waltham, MA) and imaged using a ChemiDoc MP Imaging System (Bio-Rad, Hercules, CA).

**GST pull-down assay**. Each binding experiment was comprised of 100 µL of 30 µM GST or GST-SV40Tag, combined with 100 µL of 30 µM of each importin α variant, and incubated at room temperature for 2 h with 50 µL of glutathione agarose beads (Sigma G4510). The beads were centrifuged and washed three times with 1 mL Tris wash buffer (125 mM NaCl, Tris pH 8.0). Samples were centrifuged, the supernatant discarded, and 50 µL sodium dodecyl sulfate polyacrylamide gel electrophoresis (SDS-PAGE) loading buffer containing 100 mM DTT was added to each tube. Samples were heated for 10 min at 95 °C, vortexed for 5 min, centrifuged for 10 min at 17 000 × g and analysed by SDS-PAGE (165 V for 30 min on a 4–12% Bis-Tris plus gel (Novagen)).

## Data availability

Atomic coordinates and related structure factors have been deposited to the Protein Data Bank with accession codes 6BW1, 6BW0, 6BW9, 6BWA, 6BWB, 6BVV, 6BVT and 6BVZ for the importin α1:HeV W ($P2_12_12_1$ space group), importin α1:NiV W ($P2_12_12_1$ space group), importin α3:HeV W crystal form 1 ($P12_11$ space group), importin α3:HeV W crystal form 2 ($P12_11$ space group), importin α3:HeV W crystal form 3 ($P2_12_12_1$ space group), importin α3:NiV W ($P12_11$space group), importin α1 ($P2_12_12_1$ space group), importin α3 ($P12_11$ space group), respectively. All other data that support the findings of this study are available from the corresponding authors on reasonable request.

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

## Acknowledgements

Funding from NIH grants U19AI109664 and U19AI109945 assisted this work. This research was undertaken in part using the MX2 beamline at the Australian Synchrotron, part of ANSTO, and made use of the Australian Cancer Research Foundation (ACRF) Detector. TPSC is supported by an NHMRC Fellowship (APP1091976). We also acknowledge the La Trobe University-Comprehensive Proteomics Platform for providing infrastructure. CFB is a Georgia Research Alliance Eminent Scholar in Microbial Pathogenesis. ST and EC were supported through a Graham Centre scholarship.

## Author contributions

K.M.S. performed protein expression, purification, crystallisation, structure determination, structure analysis and manuscript preparation, S.T. performed protein expression, purification, crystallisation, structure determination and manuscript preparation, M.R.E. performed cloning, pull-down and nuclear import experiments, and manuscript preparation, J.B. performed cloning, T.P.S.C. performed MST experiments, D.A. assisted with data collection and structure determination, E.M.C. performed protein expression,

purification, crystallisation, C.F.B. performed experiment design and analysis and manuscript preparation, J.K.F. performed structure determination and manuscript preparation.

## Additional information

**Competing interests:** The authors declare no competing interests.

