## [Peer Review File · Nature Communications]

Reviewers' comments:

Reviewer #1 (Remarks to the Author):

The paper "Structural basis for specificity within the importin α nuclear import receptor subfamily" by Kate Smith et al presents a structure/function analysis of the binding-specificity of the W proteins of Hendra virus (HeV) and Nipah virus (NiV) for importin α isoforms. The emphasis of this paper is on the isoform importin $\alpha 3$, which is also implicated in nuclear import of vital cellular NLS-cargos such as NF- κ B, RCC1, HIV-1 integrase, etc. Overall, the paper is well written, properly referenced and the experimental work is of excellent quality, which also includes nice and clearly illustrated figures.

My main criticism is that the primary finding of this work that ARM 7-8 of importin $\alpha 3$ are responsible for NLS-cargo specificity is not backed up by strong mutational/biochemical data. The chimeras presented in the paper are necessary, but not sufficient to pinpoint that all specificity-determinants in importin $\alpha 3$ reside in ARM 7-8. It is important to identify either loss of function mutations in ARM 7-8 of importin $\alpha 3$ that remove isoform-selectivity for HeV W, or gain of function mutations in importin $\alpha 1$ ARM 7-8 (based on importin $\alpha 3$) that confer specificity for HeV W. Either mutation would narrow down the molecular determinants for importin $\alpha 3$ specificity and make this work more mechanistic (and perhaps less circumstantial).

I have a few specific comments that I invite the authors to address.

Line 1

"Structural basis for specificity within the importin α nuclear import receptor subfamily"

The title is too broad and, in my opinion, should rather focus on the recognition of W proteins of Hendra virus (HeV) and Nipah virus (NiV) by importin $\alpha 3$.

Line 21

"There are seven human receptor importin α isoforms that mediate nuclear import of cargo in a tissue- and isoform-specific manner".

This is a bit confusing. I realize the authors may have good (both objective and subjective) reasons to refer to importin α as an import 'receptor', but the vast majority of papers in nuclear transport and books/reviews/scientists in the field refers to importin α as the import 'adaptor' of the 'receptor' importin β . Stating in the abstract that importin α is a 'receptor' makes it very confusing and doesn't help the reader, especially those readers from other fields/disciplines that are not too familiar with the nuclear import jargon.

Line 163-167

"Our structural analysis of the W protein bound to importin $\alpha 1$ and $\alpha 3$ showed that all binding determinants on importin $\alpha 3$ are conserved in importin $\alpha 1$ (Figure 5A and Extended Figure 1), suggesting that isoform specificity is not due to differences in the NLS binding groove".

This agrees well with previous data reported by Pumroy et al, Structure 2015. Citing this work here would seem appropriate.

Lines 184-186

"Similarly, superposition of a variety of importin $\alpha 3$ structures confirmed rigidity and favourable binding in this region (Figure 5D), implying that the positioning of these ARM regions across a wide range of structures are conducive to high affinity binding of W in importin $\alpha 3$, but not importin $\alpha 1$ ".

I suggest rephrasing this sentence. Superimposition of two crystal structures cannot confirm (or even inform about) 'rigidity', especially for Importin $\alpha 3$, which is by far the most flexible isoform of importin α , as revealed by MD simulations (Pumroy et al, Structure 2015) and biochemical/crystallographic studies (Sankhala et al, Nat Comms 2017).

In general, I am a bit skeptical about inferring conformational flexibility on the basis of crystal structures of importin α isoforms obtained in complex with short peptides. In the absence of full length NLS-cargos, without the domains flanking an NLS (N-terminal, in this case), and subjected to crystallization packing forces, importin α can adopt a variety of conformations, which don't necessarily inform us about the true rigidity/flexibility of this molecule.

Lines 197-199

"Superposition of importin $\alpha 3$ in the cargo free and bound forms showed that the N-terminal ARM domains 1-4 are repositioned by up to 5 Å, consistent with previous findings that a more flexible hinge region is present in importin $\alpha 3$ ".

Please cite the relevant work here: Sankhala et al, Nat Comms 2017.

Lines 212-215

"We found that a chimera comprised of the N-terminus of importin $\alpha 1$ (IBB domain and ARMs 1-5) and the C-terminus of importin $\alpha 3$ (ARMs 6-10) (importin $\alpha 1$ ARM1-5: $\alpha 3$ ARM6-10) bound W, whereas the reverse chimera comprised of importin $\alpha 3$ IBB and ARMs 1-5 and importin $\alpha 1$ ARMs 6-10 (importin $\alpha 3$ ARM1-5: $\alpha 1$ ARM6-10) did not pull down W (Figure 7B)"

I like this experiment, but I am also aware that chimeras of this nature (e.g. 50%-50% split-proteins) are often poorly folded and/or poorly behaved in solution. The IP in Fig 7B lacks positive and negative controls. The authors should show that both chimeras (importin $\alpha 1$ ARM1-5: $\alpha 3$ ARM6-10 and importin $\alpha 3$ ARM1-5: $\alpha 1$ ARM6-10) can pull-down a generic NLS-cargo and this interaction is abolished by specific mutations in importin α at critical Trps. Likewise, the Elisa assay with short peptide hardly measures binding specificity. Can this Elisa be done with the full length HeV W? Have the authors tried to crystallize the apo-chimeras? That would also be a great proof of correct folding.

Lines 219-220 and 228-229

"Overall, these results suggest that the positioning of the C-terminal ARM domains 7 and 8 are important for mediating isoform specificity of Henipavirus W proteins" and "We found that the isoform specificity was localized to the C-terminal ARM domains 7 and 8, and that the positioning of these domains was an important determinant for mediating specificity".

I don't fully understand this statement. Both chimeras have been probed in solution rather than visualized crystallographically. Thus, stating that the 'positioning' of ARMs 7-8 mediates isoform specificity is not entirely accurate and justified, especially in light of the invariant NLS-binding surface shared by all importin α isoforms. What about ARM 9-10 then? Since the IP experiment in Fig. 7 is done with chimera that also include ARM 9-10 of importin $\alpha 3$, one could argue that ARM 9-10 mediate specificity rather than ARM 7-8. Why and why not? The authors should provide more compelling data supporting their hypothesis. For instance, have they tried to crystallize the two chimeras bound to HeV W peptides? I guess that would be a start and it would also re-insure the reader that the chimeras are indeed properly folded. An alternative approach would be to look at residues in ARM 7-8 of $\alpha 3$ that differ in $\alpha 1$ and introduce these residues in $\alpha 1$. This would generate a more 'discrete' chimera of importin $\alpha 1$ that ideally gains high affinity binding to HeV W.

Lines 255-256 and 267-268

"In addition, the chimeric protein importin α 1ARM1-5: α 3ARM6-10 binds to W, whereas importin α 3ARM1-5: α 1ARM6-10 does not interact, confirming that the C-terminal domain of the importin α is the differentiating factor, rather than the N-terminus" and "We demonstrate that differences in the ARM domains in the C-terminus of importin α 3 mediate specificity".

Though this may be a new concept for the isoform α 3, the importance of C-terminal ARM repeats for recognition of non-classical cargos has already been demonstrated for importin α 5 (Nardozi et al, J Mol Biol 2010).

Lines 255-256

"However, this is unlikely to play a role in the specificity of importin α 3 for the W protein, because the difference in binding interactions occur in regions outside of the hinge region, and the α 3ARM1-5: α 1ARM6-10 chimera would have interacted with W if the hinge region was the critical region for mediating specificity".

This statement overlooks one of the strongest pieces of evidence presented in this paper, that the apo-importin α 3 is drastically different in ARM 1-3 as compared to the HeV W-bound conformation. The authors should clarify this point.

Reviewer #2 (Remarks to the Author):

One of the long-standing questions within the nucleocytoplasmic transport field is how cargos differentiate between the seven isoforms of human importin- α , which all share a highly conserved, invariant, NLS binding region. Smith and co-authors present a compelling case for an alternative mode of importin- α isoform specificity. Importin- α 3 is shown to accommodate a bipartite NLS through an open orientation of ARM repeat motifs 7-8, whereas importin- α 1 is shown to bind the same NLS sequence in a mono-partite fashion, hindered by steric clashes at the aforementioned ARMs, resulting in lower binding affinity. This observation adds to a growing appreciation within the field for the subtleties of classical nuclear import, with recent studies having identified instances where both the importin- β binding domain (Pumroy et al., 2015; Structure) and interactions with adjacent topographical features present on the cargo (Sankhala et al., 2017; Nature Comms), can play a role in isoform selection. The manuscript is presented in a logical and concise manner that is easy to follow and describes a series of 8 high quality crystal structures of both importin- α 1 and α 3 in complex with two viral NLS sequences, previously identified to exhibit isoform specificity. Additionally, the authors present a structure of importin- α 3 in isolation, which is a valuable addition to the literature. The strong structural foundation is complemented by comprehensive biochemical dissection, encompassing co-IP and ELISA experiments, with their conclusions validated extensively using both mutants and some neat importin- α chimeras. However, it is my view that whilst this study is of significant interest to the nucleocytoplasmic transport field, its impact and interest to the broader Nature Communications audience is borderline. While conducted to a uniformly high technical standard, the current manuscript does not extend the broad understanding of classical import specificity sufficiently, which is especially apparent when the results are evaluated in context of other recent publications, and therefore is more appropriate for publication in a more specialized journal.

General points

- (1) Dotted throughout the paper are several unwieldy sentences, with the behemoths between lines 54-62, 74-77 and 131-134, prominent examples.
- (2) Figure quality and comprehension is poor. The figures require the attention of the senior

author.

Figure points

Figure 1 - A) This schematic is more than is required. Only the W variant is discussed. There is no mention of the P, V or C proteins roles and differences in the manuscript. I'd be inclined to remove the middle portion and simplify the panel to focus on the NLS sequences with a domain map of just the W protein. B) There is a mixture of fonts between the diagram and table. This table is potentially extraneous and may warrant a move to the supplement. C) Requires an associated supplemental figure displaying the complete Western blots (this also applies to Figures 4B and 7B). D) Colour lines rather than shapes would be easier to make out, particularly given the small size of these charts when printed. It would also be nice to have the calculated approximate affinity values noted on the graph next to the curves.

Figures 2-3 - A) The "word-art" used throughout the structure figures are distracting. Particularly, in the case of the annotated NLS sequences, a simple courier font and bolded/coloured text would be more easily interpreted. B) I really like the idea behind these schematics. However, the image needs to be of higher resolution and during re-sizing they appear to have been 'crunched' out of perspective.

Figure 4 - C) Difficult to differentiate between the different samples based on the data point markers, colour lines may prove easier to assess at a glance. D) Labels of what is being stained within the image labeled within the figure: DAPI in blue and HA-HeV/NiV in Red? DIC images of the cells shown would also be appreciated. Currently the image displayed for the K437/438D mutant appears different in both shape and size compared with both the WT and other mutants, however without DIC this is hard to assess.

Figure 5 - B) Consider representing the helices in cylinder mode for the superpositions. The box detailing the clashes with importin- α 1 lacks context, a more detailed view showing the arrangement of the helices would help orient the reader.

Figure 6 - The top surface/cartoon view serves no purpose and should be removed.

Figure 7 - Whilst the construct boundaries for the unmodified importin- α 1 and α 3 are displayed in (A), there is no reference to them within the text, this needs to be added to the plasmids section of the methods.

Additional suggestions

(1) A superposition of all three Imp- α 3 HeV structures in a supplemental figure, with a focus on ARM 7-8, would be a nice way to utilize these confidence building validation structures and demonstrate further how invariant the localization of these repeat domains is.

(2) Throughout the manuscript individual ARM repeats are referred to as domains, they would be better described as motifs and/or repeats. The continuous NLS binding region composed of ARMS 1-10 is a domain, the individual repeats are not.

(3) This manuscript would benefit from an in vivo validation of isoform specificity on viral propagation. A tissue culture RNAi approach, coupled with a viral plaque forming measurement would be desirable.

(4) Whilst the ELISA assay provides an adequate comparative affinity measurement, a more rigorous methodology such as ITC, SPR or MST could be employed to accurately determine rate constants. These methods are standard in the field.

Reviewer #3 (Remarks to the Author):

Members of importin receptor family are responsible for the transfer of proteins from the cytoplasm to the nucleus of a cell and are important targets for the design of anti-viral and anti-cancer therapeutics. The manuscript by Smith et al presents crystal structures of W protein nuclear localization signaling regions from two closely related henipaviruses, Nipah virus (NiV) and Hendra virus (HeV), in complex with high (α 3) and low (α 1) affinity importins. The combination of

site-directed mutagenesis, chimeric a1-a3 constructs, and binding studies, allowed the development of a molecular-level rationale for how the differential positioning of C-terminal ARM-repeats of a1 and a3 importins is essential for modulating NLS specificity and affinity. Structural comparison of N- and C-terminal NLS-importin structures allows the authors suggest more general rules that dictate isoform specificity.

Overall, this seems a well-performed study with supported conclusions. The structural data appears sound and this work allows new insights into the nuclear import of virulence factors from important viral pathogens.

Comments

1. The importance of the 'rules' and biology derived from this study could be described in better detail. In particular, justification of how these experiments 'will provide important insights into understanding isoform specificity (line 268)' could be enhanced.
2. The authors show that NiV-W and HeV-W recognize a1 and a3 similarly. Can the authors comment on whether this is likely to be a conserved binding mode across the Henipavirus genus? More specifically, are these interactions likely to be conserved with other HNVs such as [non-pathogenic] Cedar virus, Gh-M74a virus, and Mojiang virus?
3. How reliable are the Kd calculations derived from the ELISA, particularly the low affinity interactions (e.g. Fig. 1D, alpha-1) that have not appeared to reach saturation?
4. Line 96: awkward language '...importin alphas and probed....'
5. The interfaces presented in Figs. 2 and 3 were difficult to interpret and compare. Can the authors better highlight the meaningful similarities and differences between the presented interfaces?

REVIEWERS' COMMENTS:

Reviewer #1 (Remarks to the Author):

The paper “Structural basis for specificity within the importin α nuclear import receptor subfamily” by Kate Smith et al presents a structure/function analysis of the binding-specificity of the W proteins of Hendra virus (HeV) and Nipah virus (NiV) for importin α isoforms. The emphasis of this paper is on the isoform importin $\alpha 3$, which is also implicated in nuclear import of vital cellular NLS-cargos such as NF- κ B, RCC1, HIV-1 integrase, etc. Overall, the paper is well written, properly referenced and the experimental work is of excellent quality, which also includes nice and clearly illustrated figures.

My main criticism is that the primary finding of this work that ARM 7-8 of importin $\alpha 3$ are responsible for NLS-cargo specificity is not backed up by strong mutational/biochemical data. The chimeras presented in the paper are necessary, but not sufficient to pinpoint that all specificity-determinants in importin $\alpha 3$ reside in ARM 7-8. It is important to identify either loss of function mutations in ARM 7-8 of importin $\alpha 3$ that remove isoform-selectivity for HeV W, or gain of function mutations in importin $\alpha 1$ ARM 7-8 (based on importin $\alpha 3$) that confer specificity for HeV W. Either mutation would narrow down the molecular determinants for importin $\alpha 3$ specificity and make this work more mechanistic (and perhaps less circumstantial).

RESPONSE: Please refer also to our response to the editor. Based on our structures we have designed loss of function mutations in ARM 7-8 of Imp α 3. We demonstrate that mutations in ARMs 7-8 results in a loss of isoform selectivity. The functionality and correct folding of this mutant (as well as the gain of function chimera Imp α proteins) has also been confirmed by testing binding to the classical SV40 T-ag NLS. This data has been incorporated into Figure 7 and Extended Figure 6.

I have a few specific comments that I invite the authors to address.

Line 1

“Structural basis for specificity within the importin α nuclear import receptor subfamily”

The title is too broad and, in my opinion, should rather focus on the recognition of W proteins of Hendra virus (HeV) and Nipah virus (NiV) by importin $\alpha 3$.

RESPONSE: We have modified the title to: “Structural basis for importin alpha 3 binding specificity of W proteins in Hendra and Nipah viruses”

Line 21

“There are seven human receptor importin α isoforms that mediate nuclear import of cargo in a tissue- and isoform-specific manner”.

This is a bit confusing. I realize the authors may have good (both objective and subjective) reasons to refer to importin α as an import ‘receptor’, but the vast majority of papers in nuclear transport and books/reviews/scientists in the field refers to importin α as the import ‘adaptor’ of the ‘receptor’ importin β . Stating in the abstract that importin α is a

'receptor' makes it very confusing and doesn't help the reader, especially those readers from other fields/disciplines that are not too familiar with the nuclear import jargon.

RESPONSE: We have modified all references to importin alpha as "adapter".

Line 163-167

"Our structural analysis of the W protein bound to importin $\alpha 1$ and $\alpha 3$ showed that all binding determinants on importin $\alpha 3$ are conserved in importin $\alpha 1$ (Figure 5A and Extended Figure 1), suggesting that isoform specificity is not due to differences in the NLS binding groove".

This agrees well with previous data reported by Pumroy et al, Structure 2015. Citing this work here would seem appropriate.

RESPONSE. We have incorporated this reference

Lines 184-186

"Similarly, superposition of a variety of importin $\alpha 3$ structures confirmed rigidity and favourable binding in this region (Figure 5D), implying that the positioning of these ARM regions across a wide range of structures are conducive to high affinity binding of W in importin $\alpha 3$, but not importin $\alpha 1$ ".

I suggest rephrasing this sentence. Superimposition of two crystal structures cannot confirm (or even inform about) 'rigidity', especially for Importin $\alpha 3$, which is by far the most flexible isoform of importin α , as revealed by MD simulations (Pumroy et al, Structure 2015) and biochemical/crystallographic studies (Sankhala et al, Nat Comms 2017).

In general, I am a bit skeptical about inferring conformational flexibility on the basis of crystal structures of importin α isoforms obtained in complex with short peptides. In the absence of full length NLS-cargos, without the domains flanking an NLS (N-terminal, in this case), and subjected to crystallization packing forces, importin α can adopt a variety of conformations, which don't necessarily inform us about the true rigidity/flexibility of this molecule.

RESPONSE. We have rephrased this section to not refer to rigidity. It now reads: Similarly, superposition of a variety of importin $\alpha 3$ structures confirmed these regions are highly similar in all structures and allow for favourable binding (Figure 5D). This supports the notion that the positioning of these ARM regions, highly similar across a wide range of structures, is conducive to high affinity binding of W in importin $\alpha 3$, but not importin $\alpha 1$.

Lines 197-199

"Superposition of importin $\alpha 3$ in the cargo free and bound forms showed that the N-terminal ARM domains 1-4 are repositioned by up 5 Å, consistent with previous findings that a more flexible hinge region is present in importin $\alpha 3$ ".

Please cite the relevant work here: Sankhala et al, Nat Comms 2017.

RESPONSE. We have included the relevant reference.

Lines 212-215

“We found that a chimera comprised of the N-terminus of importin $\alpha 1$ (IBB domain and ARMs 1-5) and the C-terminus of importin $\alpha 3$ (ARMs 6-10) (importin $\alpha 1$ ARM1-5: $\alpha 3$ ARM6-10) bound W, whereas the reverse chimera comprised of importin $\alpha 3$ IBB and ARMs 1-5 and importin $\alpha 1$ ARMs 6-10 (importin $\alpha 3$ ARM1-5: $\alpha 1$ ARM6-10) did not pull down W (Figure 7B)”

I like this experiment, but I am also aware that chimeras of this nature (e.g. 50%-50% split-proteins) are often poorly folded and/or poorly behaved in solution. The IP in Fig 7B lacks positive and negative controls. The authors should show that both chimeras (importin $\alpha 1$ ARM1-5: $\alpha 3$ ARM6-10 and importin $\alpha 3$ ARM1-5: $\alpha 1$ ARM6-10) can pull-down a generic NLS-cargo and this interaction is abolished by specific mutations in importin as at critical Trps. Likewise, the Elisa assay with short peptide hardly measures binding specificity. Can this Elisa be done with the full length HeV W? Have the authors tried to crystallize the apo-chimeras? That would also be a great proof of correct folding.

RESPONSE. To confirm the Imp α chimera functionality (and also the new Imp $\alpha 3$ mutant, see below), we have confirmed binding of importin $\alpha 1$ and $\alpha 3$ wild type, mutants, and chimeras to the classical NLS cargo SV40 T-ag (new Extended Figure 6). This result includes the gain of function chimera mutant, importin $\alpha 1$ ARM1-5: $\alpha 3$ ARM6-10. As described in the manuscript, the importin $\alpha 3$ ARM1-5: $\alpha 1$ ARM6-10 loss of function chimera could not be recombinantly expressed in *E. coli*. However, we have been able to construct and test a loss of function mutant involving critical binding residues in ARM7/8: W348A/N352A/E387A/ N394A. This mutant exhibits loss of binding (new Figure 7), and could be recombinantly expressed and purified allowing confirmation that this mutant binds the classical SV40 T-ag NLS.

ELISA cannot be done with the full length W due to its poor expression in *E. coli*. The full protein contains large disordered regions Eg: Structural disorder and modular organization in Paramyxovirinae N and P. Karlin D1, Ferron F, Canard B, Longhi S. J Gen Virol. 2003 Dec;84(Pt 12):3239-52. However, all our experiments encompass a large portion of the C-terminal region of the W protein that extends past the NLS binding region in both directions.

We have attempted to crystallise the chimera and mutant proteins. The chimera crystallised and diffracted to 8 Å; the ARM7/8 mutant crystallised and diffracted to 4 Å. This precludes detailed structural analysis, however the binding data above confirms the functional integrity of the chimera/mutants.

Lines 219-220 and 228-229

“Overall, these results suggest that the positioning of the C-terminal ARM domains 7 and 8 are important for mediating isoform specificity of Henipavirus W proteins” and “We found that the isoform specificity was localized to the C-terminal ARM domains 7 and 8, and that the positioning of these domains was an important determinant for mediating specificity”.

I don't fully understand this statement. Both chimeras have been probed in solution rather than visualized crystallographically. Thus, stating that the ‘positioning’ of ARMs 7-8 mediates isoform specificity is not entirely accurate and justified, especially in light of the invariant NLS-binding surface shared by of all importin α isoforms. What about ARM 9-10

then? Since the IP experiment in Fig. 7 is done with chimera that also include ARM 9-10 of importin $\alpha 3$, one could argue that ARM 9-10 mediate specificity rather than ARM 7-8. Why and why not? The authors should provide more compelling data supporting their hypothesis. For instance, have they tried to crystallize the two chimeras bound to HeV W peptides? I guess that would be a start and it would also re-insure the reader that the chimeras are indeed properly folded. An alternative approach would be to look at residues in ARM 7-8 of $\alpha 3$ that differ in $\alpha 1$ and introduce these residues in $\alpha 1$. This would generate a more 'discrete' chimera of importin $\alpha 1$ that ideally gains high affinity binding to HeV W.

RESPONSE. This statement was made from observations encompassing both the structural data as well as the mutant/chimera data. We agree with the reviewer that the solution data alone can lead to this conclusion. To avoid confusion, we have reworded this section as below. We have also performed the ARM7/8 mutations in $\alpha 3$, providing additional data that supports the structural and functional work.

“Overall, these results, together with the structural data, suggest that the positioning of the C-terminal ARM domains 7 and 8 are important for mediating isoform specificity of Henipavirus W proteins”

Lines 255-256 and 267-268

“In addition, the chimeric protein importin $\alpha 1$ ARM1-5: $\alpha 3$ ARM6-10 binds to W, whereas importin $\alpha 3$ ARM1-5: $\alpha 1$ ARM6-10 does not interact, confirming that the C-terminal domain of the importin α is the differentiating factor, rather than the N-terminus” and “We demonstrate that differences in the ARM domains in the C-terminus of importin $\alpha 3$ mediate specificity”.

Though this may be a new concept for the isoform $\alpha 3$, the importance of C-terminal ARM repeats for recognition of non-classical cargos has already been demonstrated for importin $\alpha 5$ (Nardozzi et al, J Mol Biol 2010).

RESPONSE. We have incorporated this reference into the manuscript.

Lines 255-256

“However, this is unlikely to play a role in the specificity of importin $\alpha 3$ for the W protein, because the difference in binding interactions occur in regions outside of the hinge region, and the $\alpha 3$ ARM1-5: $\alpha 1$ ARM6-10 chimera would have interacted with W if the hinge region was the critical region for mediating specificity”.

This statement overlooks one of the strongest pieces of evidence presented in this paper, that the apo-importin $\alpha 3$ is drastically different in ARM 1-3 as compared to the HeV W-bound conformation. The authors should clarify this point.

RESPONSE. We have incorporated additional text: “Additionally, we show through structural comparisons between apo- and W-bound-importin $\alpha 3$, that structural changes in the N-terminus are not associated with direct binding interactions in this region.”

Reviewer #2 (Remarks to the Author):

One of the long-standing questions within the nucleocytoplasmic transport field is how cargos differentiate between the seven isoforms of human importin- α , which all share a highly conserved, invariant, NLS binding region. Smith and co-authors present a compelling case for an alternative mode of importin- α isoform specificity. Importin- $\alpha 3$ is shown to accommodate a bipartite NLS through an open orientation of ARM repeat motifs 7-8, whereas importin- $\alpha 1$ is shown to bind the same NLS sequence in a mono-partite fashion, hindered by steric clashes at the aforementioned ARMs, resulting in lower binding affinity. This observation adds to a growing appreciation within the field for the subtleties of classical nuclear import, with recent studies having identified instances where both the importin- β binding domain (Pumroy et al., 2015; Structure) and interactions with adjacent topographical features present on the cargo (Sankhala et al., 2017; Nature Comms), can play a role in isoform selection. The manuscript is presented in a logical and concise manner that is easy to follow and describes a series of 8 high quality crystal structures of both importin- $\alpha 1$ and $\alpha 3$ in complex with two viral NLS sequences, previously identified to exhibit isoform specificity. Additionally, the authors present a structure of importin- $\alpha 3$ in isolation, which is a valuable addition to the literature. The strong structural foundation is complemented by comprehensive biochemical dissection, encompassing co-IP and ELISA experiments, with their conclusions validated extensively using both mutants and some neat importin- α chimeras. However, it is my view that whilst this study is of significant interest to the nucleocytoplasmic transport field, its impact and interest to the broader Nature Communications audience is borderline. While conducted to a uniformly high technical standard, the current manuscript does not extend the broad understanding of classical import specificity sufficiently, which is especially apparent when the results are evaluated in context of other recent publications, and therefore is more appropriate for publication in a more specialized journal.

General points

(1) Dotted throughout the paper are several unwieldy sentences, with the behemoths between lines 54-62, 74-77 and 131-134, prominent examples.

RESPONSE. We have revised these sentences. They now read:

For example, both RCC1 (the exchange factor of Ran that regulates the directionality of nuclear transport) and HIV-1 integrase (responsible for integrating the HIV-1 genome into the DNA of an infected cell), bind specifically to importin $\alpha 3$ ^{15,16}. STAT1, a signalling molecule in the innate immune system response, binds specifically to the convex C-terminal surface of importin $\alpha 5$, $\alpha 6$ and $\alpha 7$ ^{17,18}. The avian influenza PB2 viral polymerase subunit which is a major virulence determinant, has isoform specificity for importin $\alpha 3$ in avian hosts and importin $\alpha 7$ in mammalian hosts, providing a kinetic advantage due to lower importin α auto-inhibition by the importin beta binding domain¹⁹.

In the context of NiV infection, the non-structural W protein plays an important role in virulence²⁷⁻²⁹. It has been demonstrated to antagonize innate antiviral defences by

blocking interferon (IFN) induced gene expression and by preventing expression of type I IFNs, with nuclear localization shown to be important for the latter function^{20,30}.

We observed binding patterns that were very similar to those seen with HeV W for both importin $\alpha 1$ and $\alpha 3$ (Figure 3). Crystals of the importin $\alpha 1$:NiV W complex (that had P2₁2₁2₁ symmetry and diffracted to 2.1 Å resolution) bound residues 436-443 of the NiV W C-terminal domain. In comparison, the importin $\alpha 3$:NiV W complex had P12₁1 symmetry, diffracted to 2.3 Å resolution, and showed more extensive binding, with residues 421-446 bound to importin $\alpha 3$. The binding interface was also similar to HeV W, with the importin $\alpha 1$:NiV W complex mediated by 15 hydrogen bonds, 1 salt bridge interaction, and a buried surface area of 688.7 Å². The importin $\alpha 3$:NiV W interface was mediated through 31 hydrogen bonds, 7 salt bridge interactions, and buried 1,591.8 Å² of surface area. These results indicate that the interaction of the Henipavirus W proteins with importins is highly conserved.

(2) Figure quality and comprehension is poor. The figures require the attention of the senior author.

Figure points

Figure 1 - A) This schematic is more than is required. Only the W variant is discussed. There is no mention of the P, V or C proteins roles and differences in the manuscript. I'd be inclined to remove the middle portion and simplify the panel to focus on the NLS sequences with a domain map of just the W protein.

RESPONSE. We have incorporated this change. Please see new Figure 1. As suggested, the middle portion is removed, and the panel focuses on the W protein NLSs. We have moved panel A to the extended data to show the context of the P-gene in the viral genome.

B) There is a mixture of fonts between the diagram and table. This table is potentially extraneous and may warrant a move to the supplement.

RESPONSE. We have moved this to extended data, and incorporated a consistent font throughout this and all figures.

C) Requires an associated supplemental figure displaying the complete Western blots (this also applies to Figures 4B and 7B).

RESPONSE. We have incorporated this change in Extended Figure 2.

D) Colour lines rather than shapes would be easier to make out, particularly given the small size of these charts when printed. It would also be nice to have the calculated approximate affinity values noted on the graph next to the curves.

RESPONSE. We have incorporated this change.

Figures 2-3 - A) The "word-art" used throughout the structure figures are distracting. Particularly, in the case of the annotated NLS sequences, a simple courier font and

bolded/coloured text would be more easily interpreted. B) I really like the idea behind these schematics. However, the image needs to be of higher resolution and during re-sizing they appear to have been 'crunched' out of perspective.

RESPONSE. A) We have replaced the word art to a simple font. B) The font size and resolution have been increased

Figure 4 - C) Difficult to differentiate between the different samples based on the data point markers, colour lines may prove easier to assess at a glance.

RESPONSE. We have incorporated colour into the line and data markers.

D) Labels of what is being stained within the image labeled within the figure: DAPI in blue and HA-HeV/NiV in Red? DIC images of the cells shown would also be appreciated. Currently the image displayed for the K437/438D mutant appears different in both shape and size compared with both the WT and other mutants, however without DIC this is hard to assess.

RESPONSE. We have relabelled the images to clearly show DAPI and HeV/NiV W protein labelling. DIC images cannot be included as they were not captured at the time. We did not observe any differences in cell morphology and the full quantitation of Fn/c across 50 cells is provided in the adjacent panel.

Figure 5 - B) Consider representing the helices in cylinder mode for the superpositions. The box detailing the clashes with importin- α 1 lacks context, a more detailed view showing the arrangement of the helices would help orient the reader.

RESPONSE. We have presented helices now in cylinder mode, and added more context to the figure insert by showing the position of the helices/cylinders.

Figure 6 - The top surface/cartoon view serves no purpose and should be removed.

RESPONSE. We have removed the top panel as suggested

Figure 7 - Whilst the construct boundaries for the unmodified importin- α 1 and α 3 are displayed in (A), there is no reference to them within the text, this needs to be added to the plasmids section of the methods.

RESPONSE. We have added the text reference.

Additional suggestions

(1) A superposition of all three Imp- α 3 HeV structures in a supplemental figure, with a focus on ARM 7-8, would be a nice way to utilize these confidence building validation structures and demonstrate further how invariant the localization of these repeat domains is.

RESPONSE. We have incorporated this as new Extended Figure 3.

(2) Throughout the manuscript individual ARM repeats are referred to as domains, they would be better described as motifs and/or repeats. The continuous NLS binding region composed of ARMS 1-10 is a domain, the individual repeats are not.

RESPONSE. We have incorporated this change throughout the document

(3) This manuscript would benefit from an in vivo validation of isoform specificity on viral propagation. A tissue culture RNAi approach, coupled with a viral plaque forming measurement would be desirable.

RESPONSE. Interpretation of the suggested experiment would be complicated by the fact that other henipavirus proteins (i.e. the matrix protein (M)) are also nuclear. Nuclear translocation of NiV M appears to be required for virus assembly. Further, we cannot exclude other roles for nuclear import in henipavirus replication. Finally, W is not essential for replication in cell culture but rather appears to be critical for modulating innate immunity in vivo (Satterfield et al. Nat Comm 6:7483, 2015). Therefore, any effects of ImpA knockdown would be problematic to attribute to the W protein.

(4) Whilst the ELISA assay provides an adequate comparative affinity measurement, a more rigorous methodology such as ITC, SPR or MST could be employed to accurately determine rate constants. These methods are standard in the field.

RESPONSE. We have performed MST and incorporated this into Figure 1.

Reviewer #3 (Remarks to the Author):

Members of importin receptor family are responsible for the transfer of proteins from the cytoplasm to the nucleus of a cell and are important targets for the design of anti-viral and anti-cancer therapeutics. The manuscript by Smith et al presents crystal structures of W protein nuclear localization signaling regions from two closely related henipaviruses, Nipah virus (NiV) and Hendra virus (HeV), in complex with high (a3) and low (a1) affinity importins. The combination of site-directed mutagenesis, chimeric a1-a3 constructs, and binding studies, allowed the development of a molecular-level rationale for how the differential positioning of C-terminal ARM-repeats of a1 and a3 importins is essential for modulating NLS specificity and affinity. Structural comparison of N- and C-terminal NLS-importin structures allows the authors suggest more general rules that dictate isoform specificity.

Overall, this seems a well-performed study with supported conclusions. The structural data appears sound and this work allows new insights into the nuclear import of virulence factors from important viral pathogens.

Comments

1. The importance of the 'rules' and biology derived from this study could be described in better detail. In particular, justification of how these experiments 'will provide important insights into understanding isoform specificity (line 268)' could be enhanced.

RESPONSE. There are too few structures available to devise rules regarding which NLS will interact with which isoform. We have characterised a mechanism of isoform specificity which is distinct from only one other structural study. Thus, we believe these experiments do provide important insights into isoform specificity, but at this point, it is too early and speculative to devise general rules.

2. The authors show that NiV-W and HeV-W recognize a1 and a3 similarly. Can the authors comment on whether this is likely to be a conserved binding mode across the Henipavirus genus? More specifically, are these interactions likely to be conserved with other HNVs such as [non-pathogenic] Cedar virus, Gh-M74a virus, and Mojiang virus?

RESPONSE. The apparently non-pathogenic HNV Cedar virus is reported to not edit its P gene mRNAs and is therefore not expected to encode a W protein. Based on the reported sequence (accession NC_025352) Mojiang virus isolate Tongguan1 potentially encodes a W protein, however, the predicted W protein lacks the major site binding sequence PPTKKARV that is present in both HeV and NiV W. By our analysis of sequence (accession HQ660129) Gh-M74a virus may encode a W in which the minor site binding region has an HR rather than an RR (characteristic of HeV and NiV Ws) and would have a sequence GPAVKSKT rather than the major site binding sequence PPTKKARV that is present in both HeV and NiV W (when aligned using Multalin). Further, the putative mRNA editing site in this P gene occur downstream of that in NiV or HeV P, and therefore this W terminates much later than the W of NiV and HeV. These differences could conceivably contribute to their relative attenuation.

3. How reliable are the Kd calculations derived from the ELISA, particularly the low affinity interactions (e.g. Fig. 1D, alpha-1) that have not appeared to reach saturation?

RESPONSE. We have performed MST to complement the ELISA data (please see new Figure 1). Whilst the accuracy is low for the poor affinity interactions, they still provide a useful comparison between the different importins.

4. Line 96: awkward language ‘...importin alphas and probed...’

RESPONSE. We have reworked this sentence. To both confirm this result and assess W binding against a more extensive range of importin α isoforms, we performed immunoprecipitation assays against respective importins, and probed for the presence of HeV and NiV W.

5. The interfaces presented in Figs. 2 and 3 were difficult to interpret and compare. Can the authors better highlight the meaningful similarities and differences between the presented interfaces?

RESPONSE. We have prepared a new summary of the binding interface comparisons. This has been incorporated into the manuscript as Extended Table 3.

Reviewers' comments:

Reviewer #1 (Remarks to the Author):

The paper has been revised to address some of the questions raised by reviewers. Overall, the data presented in the paper are of high quality. The paper is well written and carefully presented. I do disagree with the interpretation of the data and on the repercussion the model presented in the discussion can have on the transport community. I still believe the authors could address my concerns by restructuring the discussions avoiding overstating the role of importin $\alpha 3$ C-terminus.

Specifically, I am still confused about a few things:

Figure 1. The legend suggests the binding assays are done using Henipavirus W proteins, but in the methods, and reading the response to reviewers, it appears all binding assays are done using bacterially expressed W peptides encompassing HeV C-terminal domain (res 409-448) and NiV res 411-450. The authors must clarify this point. Figure 1 established the binding of two peptides to DIBB importin α isoforms. No binding assay has been done using the full-length NLS-cargos.

Figure 5. The structural difference between Arm 7-8 of importin $\alpha 1$ and $\alpha 3$ is minimal. Looking at Fig. 5B, why not pointing at differences in Arm 5 or Arm 2 then? Also, where in Arm 7-8 are these differences observed? Is it in the loops connecting helices or in the position of α -helices?

Discussion. In my opinion, the data presented in this paper support the previous finding that importin $\alpha 3$ has a more flexible solenoid than other isoforms, which can stretch to accommodate a variety of NLSs. While this intrinsic motion is likely governed by domains flanking the NLS (and thus can hardly be characterized studying short peptides) the variations in RMSD in Arm 7-8 are not very significant, as well as all chimera disrupting the intrinsic flexibility of importin α solenoid fail to capture the true domain motion that is likely lost in such chimera.

Therefore I don't find the evidence presented in this paper very compelling.

Based on the nice data presented in this paper, the only thing we can say for sure is:

1. importin $\alpha 3$ more flexible solenoid promotes higher affinity binding to certain NLS by undergoing opening of Arm repeats 1-4 (as previously observed). The involvement of the Arm 7-8 in high affinity binding to HeV and NiV NLSs is not supported by strong evidence.

2. The involvement of Arm 7-8 residues in NLS-specificity is not supported by compelling evidence. Residues in importin $\alpha 3$ that make specific contacts with HeV and NiV NLSs are conserved in other importin α isoforms that bind the same NLS-peptide with low affinity. Mutating these residues in importin $\alpha 3$ and showing decrease binding affinity cannot be used as a proof of 'specificity'. Thus a model where importin $\alpha 3$ specificity is dictated by its C-terminus is likely incorrect.

Reading in the discussions that the "... C-terminal NLS binding groove of importin α is the differentiating factor" is a misrepresentation of the data presented in this work.

Reviewer #2 (Remarks to the Author):

The revised manuscript is a substantial improvement, the manuscript text is now much easier to follow. All of my direct queries have been addressed satisfactorily. Furthermore, I am especially pleased to see that the suggested MST experiments have been conducted, they bring extra weight to an already robust biochemical dissection. Nevertheless, the figures remain an issue. Overall, the figures are of lower quality than one would expect from a Nature Communications paper. The supplementary figures in particular lack labels (Figure S6) and are scruffy. Unfortunately, the presented figures remain difficult to follow. Thus, the authors should carefully consider the following points in carrying out a final minor, mainly cosmetic revision to make the manuscript

accessible to the largest possible readership. This manuscript represents a clear and timely advance in how nuclear cargoes are specifically recognized by specialized importin isoforms and with the essential minimal modifications below I would support its publication in Nature Communications.

Specific points:

Figure 1.

- A) Remove the boxes surrounding panels, they are not utilized elsewhere in the paper.
- B) The pink background of the sequences is distracting and does not print well, replace with a white background and pink outline box if necessary.
- C) Consider substituting the WCL label with Input, this is a clunky abbreviation.

Figure 2/3

- A) Change font/colour for the HeV NiV names, the W could currently be construed as part of the NLS sequence.
- B) The interaction network figure is fantastic, but noticeably lower resolution than the rest of the figure. The dotted boxes outlining the interaction schematic are scruffy and unnecessary, the ARMS have already been colour coded in line with the sequence of the NLS present in each site.

Figure 4.

- A) Remove pink backgrounds. B) The microscopy images require scale bars.

Figure 5/6

- A) In the superposition figures it is not clear which molecule is which, adjust the names above to match the colour of the protein chain shown below.
- B) The initial panel B compares importin-A1 with importin-A3. However, the subsequent panels C and D, are comparing different structures of either A1 or separately A3 with one another, thus using the same colours is counter-intuitive. Consider picking either Red or Blue for A1/A3 and different shades for differing structures of the same protein in the later panels.

Figure S1

- A) Remove boxes.
- B) pink backgrounds.

Figure S2

- A) Gels are too small, consider splitting across multiple pages.
- B) Boxes indicating the regions present in the final figure would be appreciated.

Figure S3

- A) The whole figure needs to be larger and higher resolution.
- B) There are three structures superimposed but only two are mentioned in the legend or shown in the RMSD plot.

Figure S6

- A) Include MW marker labels.
- B) Label the importin-A and GST-SV40 bands on the gel.
- C) There is a missing loading/input gel.
- D) Why is what I presume is the importin-A forming a doublet, degradation?

Reviewer #1 (Remarks to the Author):

The paper has been revised to address some of the questions raised by reviewers. Overall, the data presented in the paper are of high quality. The paper is well written and carefully presented. I do disagree with the interpretation of the data and on the repercussion the model presented in the discussion can have on the transport community. I still believe the authors could address my concerns by restructuring the discussions avoiding overstating the role of importin $\alpha 3$ C-terminus.

Specifically, I am still confused about a few things:

Figure 1. The legend suggests the binding assays are done using Henipavirus W proteins, but in the methods, and reading the response to reviewers, it appears all binding assays are done using bacterially expressed W peptides encompassing HeV C-terminal domain (res 409-448) and NiV res 411-450. The authors must clarify this point. Figure 1 established the binding of two peptides to DIBB importin α isoforms. No binding assay has been done using the full-length NLS-cargos.

RESPONSE: We have revised the figure legend to avoid any confusion. In short, co-IP binding assays were performed using full length constructs, whilst the ELISA and MST assays were performed using the same peptide regions spanning the nuclear localisation signal used in our structural approaches. The Figure 1 legend now reads:

642 **Figure 1. The NLS regions of Henipavirus W proteins bind with high affinity and**
643 **specificity to the importin $\alpha 2$ subfamily containing importin $\alpha 3$ and $\alpha 4$.** (A) The W proteins
644 contain the Soyouz module moiety (Soyouz) and PCT disordered (Paramyxo PCT) regions
645 which are conserved across paramyxovirus phosphoproteins. However, the W proteins of HeV
646 and NiV W also possess a unique C-terminus compared with other P-gene products and contain
647 an NLS that mediates translocation of W into the nucleus. (B) HeV W and NiV W interact with
648 importin $\alpha 3$ and $\alpha 4$. Co-immunoprecipitation assay performed with Flag antibody on lysates of
649 HEK293T cells expressing Flag-tagged importin $\alpha 1$, $\alpha 3$, $\alpha 4$, $\alpha 5$, $\alpha 6$, $\alpha 7$ and HA-tagged full
650 length HeV W and NiV W, as indicated. Western blots were performed for HA and Flag. WCL,
651 whole cell lysate; IP, immunoprecipitation. pCAGGS denotes empty vector control. (C) The
652 NLS region of HeV W and NiV W interact with high affinity to importin $\alpha 3$. An ELISA was
653 performed using GST-W (GST as a negative control) proteins coated on 96-well plates, and
654 binding of 6xHis-tagged importin $\alpha 1$, $\alpha 3$, $\alpha 7$ assessed using an anti-6xHis HRP antibody. (D)
655 MST assay confirms high affinity binding of importin $\alpha 3$ to the NLS region of NiV W proteins,
656 and comparatively lower affinity binding to importin $\alpha 1$ and $\alpha 7$.

We have also modified the labels in Figure 1 to incorporate “NLS”. The labels now read Hendra W NLS:Importin- α and Nipah W NLS:Importin- α .

Figure 5. The structural difference between Arm 7-8 of importin $\alpha 1$ and $\alpha 3$ is minimal. Looking at Fig. 5B, why not pointing at differences in Arm 5 or Arm 2 then? Also, where in Arm 7-8 are these differences observed? Is it in the loops connecting helices or in the position of α -helices?

Response: The clash plot in figure 5B immediately below the structures show a clear point of difference between importin $\alpha 1$ and $\alpha 3$ in ARMs 7/8 and not ARM 5 or 2. Of all the structural differences between importin $\alpha 1$ and $\alpha 3$, only those in ARMs 7/8 cause steric clashes. These clashes are numerous and significant as shown in the insert in Figure 5B; there are no clashes in ARMs 2 and 5. Additionally, there are no differences in the binding interfaces in either of these ARM domains (see also Figures 2 and 3). To make this clear, we have added additional text to figure legend 5B as below.

Fig 5B

702 **Figure 5. Structural basis for specificity of Henipavirus W binding to the importin $\alpha 3$ (A)**

703 Importin $\alpha 3$:HeV W structure coloured by conservation of amino acids across all importin α
 704 isoforms highlights a 100% conservation in the binding interface. The HeV W NLS backbone

705 (coloured green) is shown in complex with importin $\alpha 3$, with sequence colour rendering red at
 706 30%, white 60%, and blue 100% sequence identity. The figure was created in UCSF Chimera

707 using alignments of all importin α s from Clustal Omega.

708 To identify structural differences, the structures of importin $\alpha 1$:HeV W and importin $\alpha 3$:HeV W were superimposed in UCSF
 709 Chimera using MatchEnsemble, and r.m.s.d. plots were generated using MatchAlign, and

710 MatchAssess functions. α -helices are shown as cylinders; importin $\alpha 3$ is coloured orange

711 throughout, and colour conservation rendering for importin $\alpha 1$ is set for blue, less than 2.5 Å

712 variance, and red for variances above 2.5 Å. Molprobit was used to analyse clash data of

712 variance, and red for variances above 2.5 Å. Molprobability was used to analyse clash data of
 713 importin α 3:HeV W NLS superimposed to importin α 1, and all clashes >0.8 are shown as red
 714 crosses. Points of difference in the clash score between importin α 1 and α 3 are localised to
 715 ARMs 7 and 8. A detailed view of the clashes are presented in the right box, highlighting
 716 residues that clash with the W NLS due to the difference in positioning of the ARM domains 7
 717 and 8 in importin α 1 . All clashing residues are positioned on the α -helices of the ARM domains
 718 (C) Structural comparisons of importin α 1 bound to a range of cargo was examined by
 719 superimposing the α 1:HeV W NLS structure determined in this study (reference molecule,
 720 coloured yellow), with importin α 1 bound to a monopartite SV40T NLS¹² (1EJL), bipartite
 721 Bimax NLS^{59,60} (3UKW), and two domain bound structures of CAP80⁶¹ (3FEX, 3FEY)
 722 coloured according to r.m.s.d as described in (B). The positioning of the α -helices in ARM
 723 domains 7 and 8 are relatively static across all structures. (D) Structural comparisons of importin
 724 α 3 bound to a range of cargo was examined by superimposing the α 3:HeV W NLS structure
 725 determined in this study (reference molecule coloured orange) with the monopartite RanBP3⁶²
 726 (5ZXZ), the NiV W (this study), and domains of PB2¹⁹ (4UAE) and RCC1¹⁵ (5TBK) coloured
 727 according to r.m.s.d as described in (B). The positioning of the α -helices in ARM domains 7 and
 728 8 are also relatively static across all structures.

Fig 2

Fig 3

Discussion. In my opinion, the data presented in this paper support the previous finding that importin $\alpha 3$ has a more flexible solenoid than other isoforms, which can stretch to accommodate a variety of NLSs. While this intrinsic motion is likely governed by domains flanking the NLS (and thus can hardly be characterized studying short peptides) the variations in RMSD in Arm 7-8 are not very significant, as well as all chimera disrupting the intrinsic flexibility of importin α solenoid fail to capture the true domain motion that is likely lost in such chimera.

Therefore I don't find the evidence presented in this paper very compelling.

RESPONSE: The structural data presented in our paper, specifically, structures of importin $\alpha 3$ in both apo- and NLS bound forms, show that importin $\alpha 3$ is able to adopt different conformations in the N-terminal ARM domains (ARMS 1-4), while the C-terminal ARM domains have highly similar conformations over a very wide range NLS bound structures - this is shown clearly in Figures 5D (see below). Our data do not support stretching of importin $\alpha 3$ as suggested by the reviewer to accommodate a variety of NLS - this is very clear in Figure 5D showing a structural analysis of a large variety of NLSs bound to importin $\alpha 3$. It is also clear that there is no additional stretching in importin $\alpha 3$ compared to $\alpha 1$ to accommodate binding to the HeV or NiV NLSs (Figure 5B). Differences in the N-terminal ARM domains are clearly not the point of difference between binding with importin $\alpha 1$ and $\alpha 3$ as shown Figures 2 and 3 - in fact importin α binding is near identical in these N-terminal ARM domains. Overall, our data provides compelling evidence that ARMs 7-8 play a role in the isoform specificity of HeV/NiV W proteins binding to importin $\alpha 3$. This includes, high resolution structural data that clearly show significant and additional binding interfaces residing in the C-terminal ARMs of importin $\alpha 3$ (Figures 2 and 3). Detailed binding analysis shows binding in ARMs 2-4 of importin $\alpha 1$ and $\alpha 3$ are identical, but only importin $\alpha 3$ binds these NLSs in an extended region across ARM domains 6-9. Importantly, we demonstrate that the positioning of the ARM domains 7 and 8 in

importin $\alpha 1$ produce steric clashes. Our structural data is supported by mutational data and importin chimeras. The mutagenesis has been undertaken on both sides of the interaction interface. We have mutated importin $\alpha 3$ in ARMs 7 and 8 (Figures 7), as well as on residues within the W NLSs that bind the ARM domains 7 and 8 (Figure 4). Our cell-based data supports both the structural, biophysical, and co-IP data.

Fig 5D

The reviewer suggests that the intrinsic motion of importin $\alpha 3$ is governed by domains flanking the NLS. Whilst this can be true for nuclear cargo containing an N-terminal NLS, (eg RCC1, where domains following the N-terminal NLS may induce conformational changes in the N-terminal ARMS of the importins), in the case of C-terminal NLS (this study), there are no additional binding domains to influence the flexible N-terminal ARM domains of importin $\alpha 3$. This is shown clearly in our structures where in fact the final 4 residues of the W proteins do not interact at all with either importin $\alpha 1$ or $\alpha 3$ (see Figure 2 above). We present strong evidence that for the case of these Henipavirus W proteins which contain a C-terminal NLS), binding differences reside in the C-terminus of importin $\alpha 3$. Specifically, we show significant differences in the NLS binding region of W proteins to importins $\alpha 1$ and $\alpha 3$. Our data, using a large region of the C-terminus of the protein encompassing the NLS region, demonstrates significant differences in binding between importins $\alpha 1$ and $\alpha 3$. We demonstrate that these interactions are important in the context of the full length proteins using mutation and cell based assays.

In the discussion, we present a fair comparison of other models which may also account for specificity in other NLS systems including the role of other domains outside of the NLS region. Importantly, this includes flexibility, which is likely to be relevant for

importin alpha specificity in other NLSs, such as RCC1 (from the Cingolani lab). We must stress that our data, pertaining to a C-terminal NLS is clearly different from this system since there are no additional domains to bind the N-terminus of the importins; furthermore, the N-terminal ARM domain interaction interface is nearly identical in importins $\alpha 1$ and $\alpha 3$ – they are not the determinant for specificity in this system as shown by our structural data. This is further supported by our biophysical, mutational, and cell data.

As highlighted by the ongoing Nipah virus outbreak in Kerala state India (<https://www.promedmail.org/>), Nipah and Hendra viruses remain important but incompletely understood pathogens. In addition to addressing fundamental questions regarding mechanisms of protein nuclear import, our work provides new insight into the structure, trafficking and function of important virus-encoded virulence factors.

We have added additional text to our discussion to assist with some of the reviewers concerns. These are highlighted below:

241 **Discussion**

242 In this study, we present high-resolution structures of importin α isoform adaptors bound to the
243 NLS regions of HeV W and NiV W, providing new insights into the molecular basis of importin
244 α isoform specificity. We found that HeV W and NiV W proteins bind importin $\alpha 3$
245 preferentially, and that these NLS regions interact with higher affinity over other importin α SFs.
246 Using structural approaches, we identified key features that account for adaptor specificity. We
251 found that the isoform specificity was localized to the C-terminal ARM repeats 7 and 8, and that
252 the positioning of these domains was an important determinant for mediating specificity.

253

254 Although numerous studies have reported specificity of nuclear adaptors for a wide range of
255 cargo^{19,34-37} and associated function of these interactions in many diseases³⁶, the mechanism(s)
256 behind NLS adaptor specificity has remained elusive. Recently, one study described the
257 specificity of the RCC1 factor for importin $\alpha 3$, highlighting that additional residues outside the

258 NLS were important for maintaining specificity¹⁵. A comparison between the mechanism
259 presented in the RCC1 study and our study, highlights a number of important differences. The
260 study by Sankhala et al¹⁵, indicated that the NLS of RCC1 binds in the major binding site of
261 ARMS 2-4 of both importin $\alpha 1$ and $\alpha 3$, with additional interactions occurring at the N-terminus
262 of importin $\alpha 3$ ARM repeats 1-4 and the β -propeller region of RCC1, mediated by flexibility and
263 rotation of importin $\alpha 3$ in ARM repeats 1-2. This mechanism of isoform specificity is well suited
264 to cargo containing NLSs at the N-terminus, and our structural data of the unbound and NLS
265 bound forms of importin $\alpha 3$ supports this model. This mechanism is distinct however from that
266 governing specificity for cargo such as the W proteins in this study, where the NLSs are located
267 at the C-terminus because the protein interface would not extend past the N-terminus of ARM
268 repeats 2-4. Thus, it may be possible that isoform specificity of cargo containing N-terminal
269 NLSs may reside through differential interaction of the N-terminal ARM repeats, whereas cargo
270 bearing C-terminal NLSs may show specificity through differential interaction with the C-
271 terminus of importins, as demonstrated in this study. Although further work will be needed to
272 establish how extensive these “rules” are, it is likely that different mechanisms may govern
273 isoform specificity, dependent upon the position of the NLS within the cargo.

Based on the nice data presented in this paper, the only thing we can say for sure is:

1. importin $\alpha 3$ more flexible solenoid promotes higher affinity binding to certain NLS by undergoing opening of Arm repeats 1-4 (as previously observed). The involvement of the Arm 7-8 in high affinity binding to HeV and NiV NLSs is not supported by strong evidence.

RESPONSE: As stated above, the ARM domains 1-4 are not the site of difference in this system. In fact, the conformations of importin $\alpha 1$ and $\alpha 3$ when bound to the NLSs are almost identical in this region (see figure 5B above). The site of difference is clearly in the C-terminal region (figure 2 and 3) and is supported by structural data and range of complementary assays.

2. The involvement of Arm 7-8 residues in NLS-specificity is not supported by compelling evidence. Residues in importin $\alpha 3$ that make specific contacts with HeV and NiV NLSs are conserved in other importin α isoforms that bind the same NLS-peptide with low affinity. Mutating these residues in importin $\alpha 3$ and showing decrease binding affinity cannot be used as a proof of ‘specificity’. Thus a model where importin $\alpha 3$ specificity is dictated by its C-terminus is likely incorrect.

Reading in the discussions that the ‘... C-terminal NLS binding groove of importin α is the differentiating factor’ is a misrepresentation of the data presented in this work.

RESPONSE: We are not claiming that specific residues in the ARM domains are the basis for the specificity. Rather, we show that the structural positioning of these ARM domains are important for specificity. The mutations carried out on both sides of the interface (importin α including ARMs 7 and 8; and W NLS region that bind to ARMs 7 and 8) support our structural and cellular observations.

To avoid confusion, we have modified the text accordingly:

275 The findings of our study are distinct from mechanisms that have been previously hypothesized.
276 The IBB domain has been shown to auto-inhibit differentially across isoforms¹⁹. However, our
277 results identify clear differences in the structural positioning of the C-terminal ARM domains 7-
278 8 of importin $\alpha 1$ and $\alpha 3$, accompanied with a significant increase in the binding interface at this
279 region in importin $\alpha 3$. In addition, the chimeric protein importin $\alpha 1^{\text{ARM1-5}};\alpha 3^{\text{ARM6-10}}$ binds to W,
280 whereas importin $\alpha 3^{\text{ARM1-5}};\alpha 1^{\text{ARM6-10}}$ does not interact, confirming that the C-terminal domain
281 of the importin α is the differentiating factor, rather than the N-terminus. The importance of the

Reviewer #2 (Remarks to the Author):

The revised manuscript is a substantial improvement, the manuscript text is now much easier to follow. All of my direct queries have been addressed satisfactorily. Furthermore, I am especially pleased to see that the suggested MST experiments have been conducted, they bring extra weight to an already robust biochemical dissection. Nevertheless, the figures remain an issue. Overall, the figures are of lower quality than one would expect from a Nature Communications paper. The supplementary figures in particular lack labels (Figure S6) and are scruffy. Unfortunately, the presented figures remain difficult to follow. Thus, the authors should carefully consider the following points in carrying out a final minor, mainly cosmetic revision to make the manuscript accessible to the largest possible readership. This manuscript represents a clear and timely advance in how nuclear cargoes are specifically recognized by specialized importin isoforms and with the essential minimal modifications below I would support its publication in Nature Communications.

Specific points:

Figure 1.

A) Remove the boxes surrounding panels, they are not utilized elsewhere in the paper.

RESPONSE: This has been changed as per reviewer's request

B) The pink background of the sequences is distracting and does not print well, replace with a white background and pink outline box if necessary.

RESPONSE: This has been changed as per reviewer's request

C) Consider substituting the WCL label with Input, this is a clunky abbreviation.

RESPONSE: This has been changed as per reviewer's request

Please see the revised Figure:

Figure 2/3

A) Change font/colour for the HeV NiV names, the W could currently be construed as part of the NLS sequence.

RESPONSE: This has been changed as per reviewer's request (please see pictures in reviewer 1 section)

B) The interaction network figure is fantastic, but noticeably lower resolution than the rest of the figure. The dotted boxes outlining the interaction schematic are scruffy and unnecessary, the ARMS have already been colour coded in line with the sequence of the NLS present in each site.

RESPONSE: This has been changed as per reviewer's request. Please refer to the high resolution picture uploaded

Figure 4.

A) Remove pink backgrounds. B) The microscopy images require scale bars.

RESPONSE: This has been changed as per reviewer's request. Please refer to the high resolution picture uploaded

Figure 5/6

A) In the superposition figures it is not clear which molecule is which, adjust the names above to match the colour of the protein chain shown below.

RESPONSE: The colour of the reference importins α1 and α3 are now coloured yellow and orange respectively, while the colour of r.m.s.d changes accordingly from blue to

red as indicated in the figure legends. This allows clear distinction between the two molecules.

B) The initial panel B compares importin-A1 with importin-A3. However, the subsequent panels C and D, are comparing different structures of either A1 or separately A3 with one another, thus using the same colours is counter-intuitive. Consider picking either Red or Blue for A1/A3 and different shades for differing structures of the same protein in the later panels.

RESPONSE: As above, we have chosen different colours for the importin α 1 (yellow) and α 3 (orange). The colouring and shading of the comparison models is according to rmsd. We have maintained this colouring throughout so that comparisons can be made across the different figures and panels. The figure legends have been modified accordingly.

Figure S1

A) Remove boxes.

RESPONSE: This has been changed as per reviewer's request

B) pink backgrounds.

RESPONSE: This has been changed as per reviewer's request

Figure S2

A) Gels are too small, consider splitting across multiple pages.

RESPONSE: This has been changed as per reviewer's request

B) Boxes indicating the regions present in the final figure would be appreciated.

RESPONSE: This has been incorporated as per reviewer's request

Figure S3

A) The whole figure needs to be larger and higher resolution.

RESPONSE: This has been changed as per reviewer's request

B) There are three structures superimposed but only two are mentioned in the legend or shown in the RMSD plot.

RESPONSE: Three structures are superimposed, however, one is chosen as a reference model (crystal form 1). Therefore, there are only two shown in the r.m.s.d plot. We have made this more obvious in the figure by changing the figure legend on the scale to read crystal form 1.

Figure S6

A) Include MW marker labels.

RESPONSE: This has been incorporated as per reviewer's request. We have also removed the cut gel and instead just presented the uncropped gel. There is little need to present both on the same figure.

B) Label the importin-A and GST-SV40 bands on the gel.

RESPONSE: This has been done as per reviewer's request

C) There is a missing loading/input gel.

RESPONSE: We have included an input gel below the main gel

D) Why is what I presume is the importin-A forming a doublet, degradation?

RESPONSE: This is likely a very minor degradation.